# HIF1α-regulated glycolysis promotes activation-induced cell death and IFN-γ induction in hypoxic T cells

Hongxing Shen [1], Oluwagbemiga A. Ojo [1], Haitao Ding [1], Logan J. Mullen[2], Chuan Xing[1], M. Iqbal Hossain[3], Abdelrahman Yassin[1], Vivian Y. Shi[1], Zach Lewis [1], Ewa Podgorska [1], Shaida A. Andrabi[3], Maciek R. Antoniewicz[4], James A. Bonner [1,5] & Lewis Zhichang Shi [1,3,5,6] ✉

Hypoxia is a common feature in various pathophysiological contexts, including tumor microenvironment, and IFN-γ is instrumental for anti-tumor immunity. HIF1α has long been known as a primary regulator of cellular adaptive responses to hypoxia, but its role in IFN-γ induction in hypoxic T cells is unknown. Here, we show that the HIF1α-glycolysis axis controls IFN-γ induction in both human and mouse T cells, activated under hypoxia. Specific deletion of HIF1α in T cells (*Hif1α⁻/⁻*) and glycolytic inhibition suppresses IFN-γ induction. Conversely, HIF1α stabilization by hypoxia and VHL deletion in T cells (*Vhl⁻/⁻*) increases IFN-γ production. Hypoxic *Hif1α⁻/⁻* T cells are less able to kill tumor cells in vitro, and tumor-bearing *Hif1α⁻/⁻* mice are not responsive to immune checkpoint blockade (ICB) therapy in vivo. Mechanistically, loss of HIF1α greatly diminishes glycolytic activity in hypoxic T cells, resulting in depleted intracellular acetyl-CoA and attenuated activation-induced cell death (AICD). Restoration of intracellular acetyl-CoA by acetate supplementation re-engages AICD, rescuing IFN-γ production in hypoxic *Hif1α⁻/⁻* T cells and re-sensitizing *Hif1α⁻/⁻* tumor-bearing mice to ICB. In summary, we identify HIF1α-regulated glycolysis as a key metabolic control of IFN-γ production in hypoxic T cells and ICB response.

Naïve T cells, upon receiving signals from T cell receptor (TCR) (signal 1), co-stimulatory receptor (e.g., CD28, signal 2), and various cytokines (signal 3), are activated and differentiate into distinct sub-lineages such as IFN-γ-producing $T_H1$[1], IL-17-producing $T_H17$[2], FoxP3⁺ regulatory T cells ($T_{reg}$)[3], as well as other recently-defined subsets[4]. Notwithstanding the essential role of these immune signals in determining T cell fates, accumulating evidence indicates that at the fundamental level, it is the cellular metabolism that dictates T cell activation and differentiation[5–17]. Interestingly, cellular metabolism and related

transcriptional factors play a rather selective but not universal role in T cell fate decisions[4–6,9–15]. For instance, Myc but not HIF1α controls the metabolic reprogramming in activated T cells under normoxia, switching from fatty acid/pyruvate oxidation via the TCA cycle to the glycolytic, pentose-phosphate, and glutaminolytic pathways[8], even though both are master regulators of cellular metabolism downstream of mTOR[18]. On the other hand, HIF1α-glycolysis orchestrates a metabolic checkpoint in $T_H17$ but not $T_H1$ differentiation, despite their shared augmentation of glycolytic activities[7]. While this could be due

[1]Department of Radiation Oncology, Heersink School of Medicine, University of Alabama at Birmingham (UAB-SOM), Birmingham, AL, USA. [2]Genomics Core Laboratory, Institute of Arctic Biology, University of Alaska Fairbanks, Fairbanks, Alaska, USA. [3]Department of Pharmacology and Toxicology, UAB-SOM, Birmingham, AL, USA. [4]Department of Chemical Engineering, University of Michigan, Ann Arbor, MI, USA. [5]O'Neal Comprehensive Cancer Center, UAB-SOM, Birmingham, AL, USA. [6]Department of Microbiology and Immunology Institute, UAB-SOM, Birmingham, AL, USA. ✉e-mail: Lewisshi@uabmc.edu

in part to a greater HIF1α induction in $T_H17$ than $T_H1$ cells[7], it is noteworthy to mention that those studies[7,19] were conducted under the regular cell culture condition (21% $O_2$), which is different from commonly encountered hypoxic environments in vivo.

Hypoxia is a prominent feature in various physiological and pathological settings. Physiologically, other than the tissues directly exposed to inhaled atmospheric air (such as the upper airways), most healthy tissues experience some degree of $O_2$ deprivation. For example, the $O_2$ tension in arterial blood is about ~14%[20], which is reduced to 5-6% in the interstitial space[21]. In lymphoid tissues like spleen, it is mostly ~3–4%[22] but lower in the germinal center[23]. In the gastrointestinal (GI) tract that hosts ~70–80% of the total lymphocytes in our body[24], a wide range of $O_2$ tensions exists, being almost anoxic in the lumen, where many obligate anaerobic commensal bacteria reside, and slightly higher at the base of the villi[25]. The intestinal tissue, including the lamina propria where many T cells are found, has an $O_2$ level of ~7%[26]. Pathologically, tumor microenvironment (TME) of solid tumors is known to be highly hypoxic ( ~1% $O_2$), resulting from abnormal vasculature, heightened metabolic activities of tumor cells, and other factors[27]. Likewise, severe hypoxia is detected in the inflammatory sites, due to edema, vasculitis, vasoconstriction (limiting oxygen delivery), and recruitment of polymorphonuclear cells that consume high amounts of $O_2$[28].

Hypoxia-inducible factor (HIF) 1 α subunit (HIF1α) has long been regarded as the primary regulator of cellular adaptive responses to hypoxia[29]. When $O_2$ tensions are low, hydroxylation of the prolyl residues of HIF1α by prolyl hydroxylase domain (PHD) enzymes is inhibited, preventing recognition and ubiquitination of HIF1α by the von Hippel-Lindau protein (VHL)[30,31]. In addition, hypoxia inhibits hydroxylation of an asparagine residue within C-terminal transcriptional activation domain (C-TAD) of HIF1α, facilitating its interactions with other transcriptional coactivators[32]. These modifications stabilize HIF1α, which subsequently translocate to the nucleus to regulate gene expression via transcriptional and epigenetic mechanisms. However, despite the ubiquitous distribution of hypoxia and the pivotal role of IFN-γ in immunity against intracellular pathogens and tumors[33], whether and how HIF1α-glycolysis controls IFN-γ induction in hypoxic T cells remain unknown.

Here, we find that HIF1α-glycolysis is indispensable for IFN-γ induction in hypoxic T cells. HIF1α deletion and glycolytic inhibition drastically reduces IFN-γ induction, whereas HIF1α stabilization greatly augments IFN-γ production. The HIF1α-glycolysis pathway, by sustaining [acetyl-CoA]) and AICD, governs T cell effector functions and ICB response. Consequently, acetate supplementation, by restoring [acetyl-CoA] and AICD in $Hif1α^{-/-}$ T cells, resensitizes $Hif1α^{-/-}$ tumor-bearing mice to ICB therapy. Since acetate supplementation (GTA: Glycerol Triacetate) is approved by the FDA, GTA may offer an effective strategy to overcome ICB resistance, a pressing unmet medical need.

## Results

### HIF1α but not HIF2α controls IFN-γ induction in hypoxic T cells

Two early studies using human[34] and mouse non-$T_{reg}$ ($CD4^+CD25^-$) T cells[35] showed that hypoxia and HIF1α inhibit IFN-γ production. Of note, $CD4^+CD25^-$ T cells contain ~40-50% activated effector/central memory T cells ($CD44^+$)[36,37] that are equipped to produce IFN-γ, preventing an unequivocal assessment of the role of HIF1α and hypoxia in IFN-γ induction in naïve T cells, upon activation. To address this, naïve $CD4^+$ T cells isolated from mice with specific HIF1α deletion in T cells (hereafter, $Hif1α^{-/-}$)[7] and wild-type littermate controls (WT) were activated with plate-bound anti-CD3/CD28 plus IL-2, in the presence and absence of IL-12. Cells were cultured under both normal cell culture condition (normoxia, 21% $O_2$) and hypoxic condition (hypoxia, 1% $O_2$ mimicking the hypoxic TME of solid tumors[27] and inflammatory sites[28]). While supplementation of IL-12, as expected, induced greater

amount of IFN-γ, nevertheless, IFN-γ can be robustly induced without IL-12 (Fig. S1A). Given this and to limit the confounding effects from IL-12 signaling, we primarily focused on the condition without IL-12.

In contrast to previous reports showing a negative role of HIF1α and hypoxia in IFN-γ production in $CD4^+CD25^-$ T cells[35], we found that HIF1α was essential for IFN-γ induction in naïve T cells activated under hypoxia but not normoxia (Fig. 1A), the latter of which is consistent with previous findings of a non-essential role of HIF1α in $T_H1$ differentiation under normoxia[7,19]. This differential role of HIF1α in IFN-γ induction in normoxic versus hypoxic T cells paralleled a selective downregulation of the master transcriptional regulator of IFN-γ, T-bet in hypoxic $Hif1α^{-/-}$ T cells (Fig. S1B). To rule out the possibility that reduced IFN-γ production in hypoxic $Hif1α^{-/-}$ T cells was due to diversion to other T cell lineages, we evaluated master transcription factors for $T_H2$ (Gata-3), $T_H17$ (RORγt), and $T_{reg}$ (FoxP3) (Fig. S1C), which remained unaltered. IFN-γ production by T cells was significantly increased by hypoxia compared to normoxia, likely an outcome of HIF1α stabilization and gained HIF1α function (Fig. 1B), as HIF1α deletion completely abolished this increase (Fig. 1A). Since another hypoxia inducible transcription factor HIF2α was also stabilized by hypoxia (Fig. 2B), we wanted to rule out its role in IFN-γ induction by knocking down HIF2α (Fig. S1D), which neither impaired IFN-γ induction nor abrogated hypoxia-driven increases of IFN-γ production (Fig. 1C). These results indicate that HIF1α but not HIF2α controls IFN-γ induction in hypoxic $CD4^+$ T cells.

We further confirmed the important role of HIF1α in IFN-γ induction in human $CD4^+$ T cells (Fig. 1D) as well as mouse $CD8^+$ T cells (Fig. 1E). To extend our findings to other hypoxic conditions, we performed experiments under 2.5% $O_2$ and observed similar IFN-γ reduction in $Hif1α^{-/-}$ T cells (Fig. 1F), indicating a rather universal requirement of HIF1α in IFN-γ induction under different hypoxic conditions. To gain a more complete picture of how HIF1α modulated effector functions of hypoxic T cells, we analyzed perforin (Prf) and granzyme (GzmB), two widely regarded late effector cytokines, and found they were also reduced in hypoxic $Hif1α^{-/-}$ T cells (Fig. S1E). Unlike IFN-γ associated with late stage of T cell activation (Figs. 1G and S1F) (late cytokines), IL-2 and TNF are linked to early stage of T cell activation[38,39] (early cytokines). We therefore analyzed their production. Surprisingly, hypoxic $Hif1α^{-/-}$ T cells had greater production of both IL-2 and TNF (Fig. S1G and S1H), suggesting a reciprocal regulation of late versus early cytokines by HIF1α. Of note, despite the increase of TNF production, the frequency of dual producers of IFN-γ and TNF was reduced in hypoxic $Hif1α^{-/-}$ T cells, due to the severe reduction of IFN-γ. These results supported that IFN-γ reduction in hypoxic $Hif1α^{-/-}$ T cells was not a result of global suppression of protein translation associated with the starvation of energy source. To illustrate this further, we replaced 50% of "old spent" medium with "fresh nutritious" medium daily from Days 1-4 and analyzed IFN-γ production on Day 5, IFN-γ reduction in $Hif1α^{-/-}$ T cells persisted (Fig. 1H). Together, these results argued that impaired IFN-γ induction in hypoxic $Hif1α^{-/-}$ T cells was unlikely mediated by extrinsic factors but more of a T cell-intrinsic effect. To test this, we adopted an in vitro chimeric system by mixing naïve WT T cells ($CD45.1^+$) equally with naïve WT or $Hif1α^{-/-}$ ($CD45.2^+$) T cells. Clearly, introduction of the feeders of extrinsic WT (CD45.1) factors did not correct the reduced IFN-γ production in $Hif1α^{-/-}$ T cells (Fig. 1I), confirming this is a cell-autonomous phenotype.

With the increased IFN-γ production by hypoxia-driven gain-of-function (GOF) of HIF1α, we asked how the production of other cytokines (i.e., Prf, GzmB, and IL-2) was affected. As shown in Fig. S1I, opposing the effects of HIF1α deletion in T cells (loss-of-function: LOF), Prf and GzmB were increased, while IL-2 was reduced by hypoxia. We then employed a genetic GOF approach by generating genetic mice with specific deletion of VHL in T cells (hereafter, $Vhl^{-/-}$), a negative regulator of HIF1α[30,31]. Isolated naïve WT and $Vhl^{-/-}$ T cells were

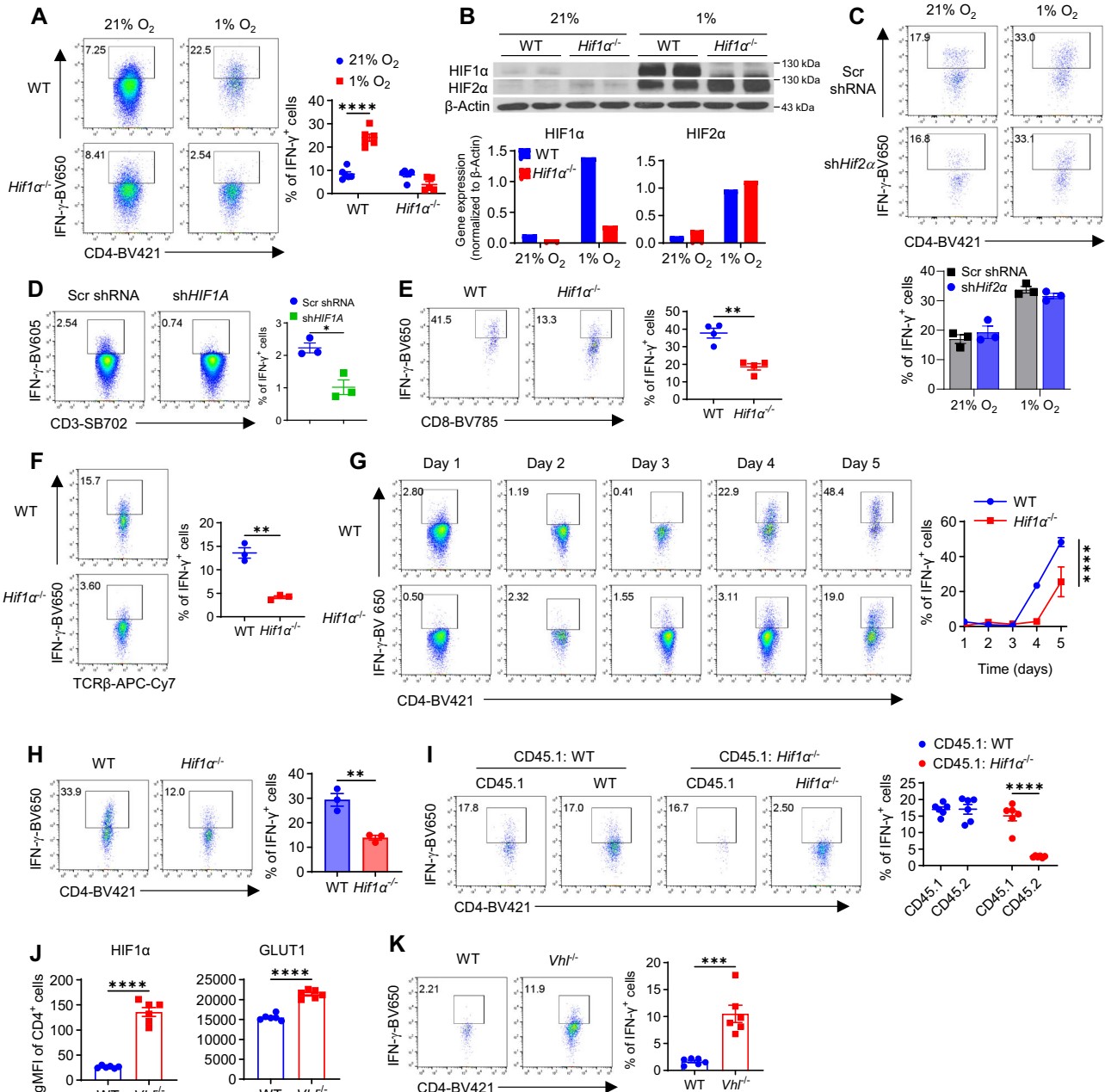

**Fig. 1 | HIF1α but not HIF2α controls IFN-γ induction in hypoxic T cells, in vitro.**
**A, B** Naïve CD4⁺ T cells isolated from WT and *Hif1α⁻/⁻* mice were activated under hypoxia (1% O₂) and normoxia (21% O₂). On Day 5, cells were harvested to measure IFN-γ production by flow cytometry. $N = 5$ for the *Hif1α⁻/⁻* 1% O₂ group, $N = 6$ for other groups (****$p < 0.0001$) (**A**). At 48 h, cell lysates were prepared to detect HIF1α and HIF2α expression by Western blot. β-Actin was included as a loading control (**B**). **C** Activated CD4⁺ T cells were transduced with retroviruses expressing scrambled shRNA (Scr shRNA) or *Hif2α* shRNA (*shHif2α*), followed by IFN-γ detection. **D** Naive human CD4⁺ T from healthy donors were activated, transduced with retroviruses expressing scrambled shRNA (Scr shRNA) or human sh*HIF1A*, cultured under hypoxia for 5 days, and detected IFN-γ production (*$p = 0.011$). **E** Naïve CD8⁺ T cells isolated from WT and *Hif1α⁻/⁻* mice were activated under hypoxia (1% O₂) for 5 days, followed by IFN-γ detection ($N = 4$, **$p = 0.0011$). **F** Naïve T cells were similarly activated as in (**A**) but under 2.5% O₂ to detect IFN-γ production ($N = 3$, **$p = 0.0012$). **G** IFN-γ production in cells from (**A**)

was evaluated every day from Day 1-5. ($N = 3$, ****$p < 0.0001$). **H** The cells were activated in hypoxia as in (**A**) but with 50% of old media replaced by fresh media daily on Day 1–4. IFN-γ production was detected on Day 5 (**$p = 0.005$). **I** Equally mixed naïve CD45.1⁺ CD4⁺ T cells with naïve CD45.2⁺ CD4⁺ WT or *Hif1α⁻/⁻* T cells were activated under hypoxia for 5 days and measured IFN-γ production ($N = 6$, ****$p < 0.0001$). **J** Geometric mean fluorescence intensity (gMFI) of HIF1α (left) and GLUT1 (right) in WT or *Vhl⁻/⁻* CD4⁺ T cells activated under hypoxia as in (**A**) ($N = 6$, ****$p < 0.0001$). **K** IFN-γ production by WT or *Vhl⁻/⁻* CD4⁺ T cells activated under hypoxia for 3 days ($N = 6$, ***$p = 0.0003$). A two-sided Student's t-test was used in **D**–**F**, **H**, **J** and **K** for statistical analyzes. Two-way ANOVA with Šídák's multiple comparisons test (with adjustment) was used for (**A**, **G**, and **I**). All the experiments were repeated at least twice. Pooled results shown in the dot plots and bar graphs depicted means ± SEM for all samples in each group, with each dot denoting an independent sample. Source data were provided in the Source Data file.

similarly activated under hypoxia. As expected, VHL deletion led to significantly increased HIF1α expression as well as its downstream target Glut1 (Fig. 1J). Importantly, *Vhl⁻/⁻* T cells produced greater amount of IFN-γ, as early as Day 2 when minimal IFN-γ production was

detected in WT T cells (Fig. 1K). There were also significant increases of Prf and GzmB in *Vhl⁻/⁻* T cells (Fig. S1J). Collectively, these congruent findings from LOF and GOF approaches compellingly establish HIF1α as a key regulator of IFN-γ induction in hypoxic T cells.

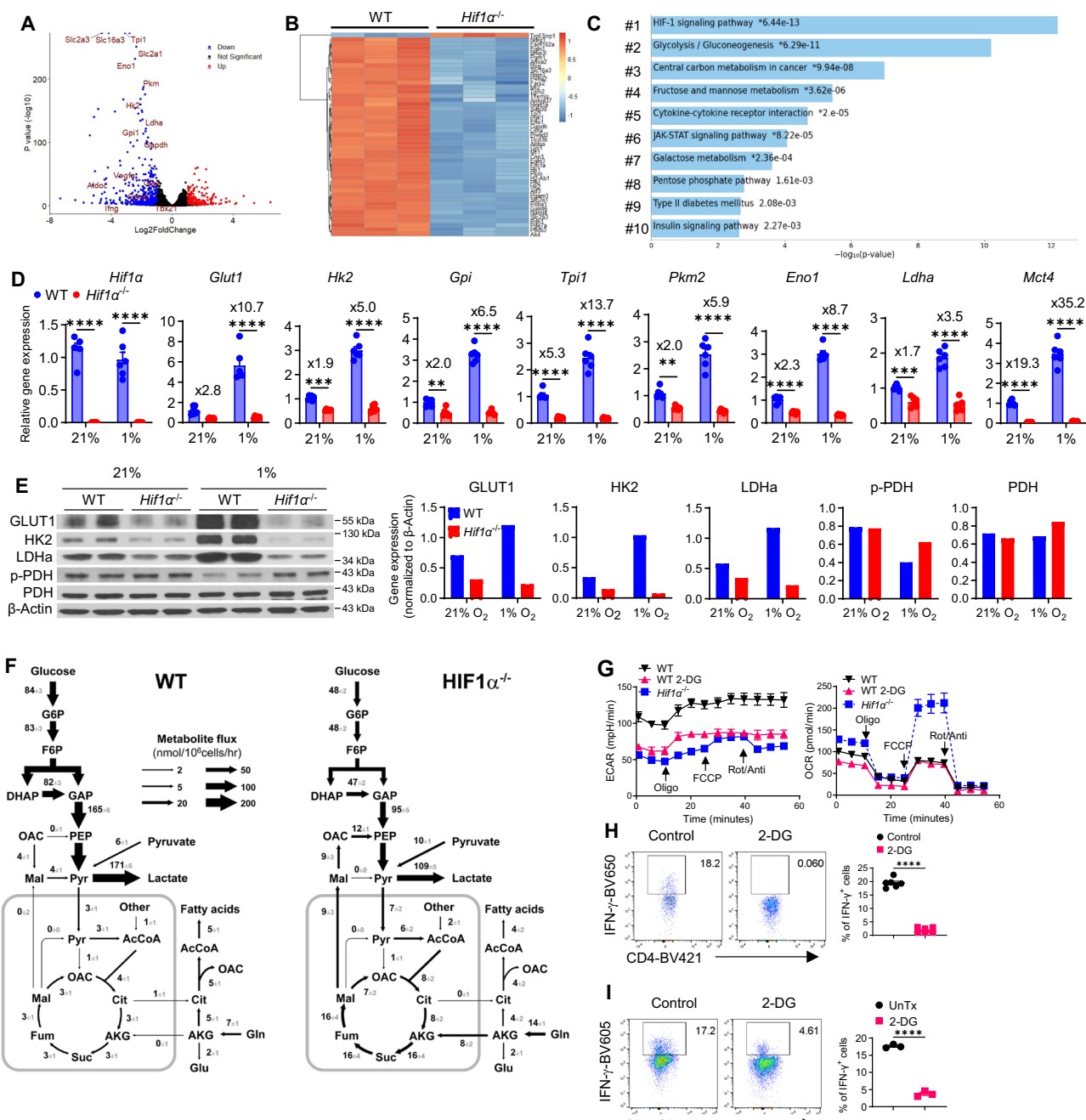

**Fig. 2 | HIF1α-glycolysis drives IFN-γ induction in hypoxic T cells. A–C** Total RNAs extracted from WT and *Hif1α*⁻/⁻ CD4⁺ naïve T cells activated under hypoxia for 48 h were subjected to RNA-Seq. The gene expression analyzes were performed using DESeq2 (version 1.34.0). The Wald test was used to calculate the *p* values and log2 fold changes. Genes with an adjusted *p* value < 0.05 and absolute log2 fold change > 1 were considered as differentially expressed genes (DEGs). A volcano plot was used to show all upregulated and downregulated DEGs using the ggplot2 R package (**A**), with top 50 identified DEGs shown as a heatmap (**B**). Top 10 enriched signaling pathways (downregulated) from Enriched Kyoto Encyclopedia of Genes and Genomes (KEGG) analyzes of DEGs were shown in (**C**). Significant terms of the KEGG pathways were selected with *p* value < 0.05. **D** mRNA expression of glycolytic genes was evaluated by real-time RT-PCR in T cells activated and cultured under normoxia (21%) and hypoxia (1%) for 48 h (*N* = 6, **p* = 0.0029 for *Gpi*, **p* = 0.0053 for *Pkm2*, ***p* = 0.0007 for *Hk2*, ***p* = 0.0009 for *Ldha*, and ****p* < 0.0001 for *Mct4* by Two-way ANOVA with Šídák's multiple comparisons test (with adjustment)). **E** Protein expression of GLUT1, HK2, LDHa, phospho-PDH and PDH was analyzed

using cell lysates prepared using cells activated as in (**D**). β-Actin was used as a loading control. **F** Metabolic flux in activated WT and *Hif1α*⁻/⁻ CD4⁺ as in (**A**), traced by ¹³C labeled glucose. The thickness of arrows relatively indicates the flow rates, with the specific numbers by the arrows depicting the metabolic flux of reactions. **G** Extracellular acidification rate (ECAR) and oxygen consumption rate (OCR) were measured using a Mito stress test kit (*N* = 4) (Oligo: Oligomycin; FCCP: carbonyl cyanide 4-(trifluoromethoxy)phenylhydrazone; Rot/Anti: Rotenone & Antimycin A). **H, I** Naïve WT mouse CD4⁺ T cells (**H**, *N* = 6 per group) and naive human CD4⁺ T cells isolated from PBMCs of healthy donors (**I**, *N* = 3 per group) were activated under hypoxia, in the presence of solvent (Control) or 0.5 μM of 2-DG for 5 days, followed by analysis of IFN-γ production. A two-sided Student's t-test was used in (**H** and **I**) for statistical analyzes (****p* < 0.0001). All the experiments were repeated at least twice. Pooled results shown in the dot plots and bar graphs depicted means ± SEM for all samples in each group, with each dot denoting an independent sample. Source data were provided in the Source Data file.

## HIF1α-orchestrated glycolysis governs IFN-γ induction in hypoxic T cells

To unravel how T cell HIF1α regulates IFN-γ expression, we conducted whole transcriptome analysis (RNA-seq) using WT and *Hif1α*$^{-/-}$ T cells activated under normoxia and hypoxia for 48 h. We chose this time-point, because it immediately preceded IFN-γ induction on Day 3-5 following T cell activation, allowing us to establish a temporal relationship of HIF1α-mediated transcriptional and metabolic changes to IFN-γ induction. Also, activated T cells rapidly died around 56 h through AICD (discussed later), and 48 h would maximize cell yield for downstream biochemical analyzes. Whereas normoxic *Hif1α*$^{-/-}$ T cells displayed limited transcriptomic changes (600 hits) with 37 well-defined differentially expressed genes (DEG) upregulated and 55 DEGs downregulated (Fig. S2A), in sharp contrast, there were 5168 hits in hypoxic *Hif1α*$^{-/-}$ T cells, among which 583 were upregulated DEGs and 399 downregulated DEGs (Fig. 2A), pointing to a substantially more prominent role of HIF1α in orchestrating transcriptomic programs in hypoxic T cells. Notably, very little overlap existed among DEGs in *Hif1α*$^{-/-}$ hypoxic versus normoxic T cells, with top 50 genes shown in heatmaps (Fig. 2B versus Fig. S2B), supporting a distinct role of HIF1α in hypoxic T cells compared to normoxic T cells. Additional signaling pathway enrichment analyzes based on downregulated DEGs also revealed minimal overlap (extended datasets provided as Supplementary Data 1-2). Among the top 10 enriched downregulated pathways, HIF1α signaling pathway was the only one shared between hypoxic (Fig. 2C) and normoxic (Fig. S2C) T cells. Likewise, barely any overlap among top upregulated enriched pathways existed between normoxic and hypoxic *Hif1α*$^{-/-}$ T cells (Fig. S2D and Supplementary Data 1-2). A closer look at the other 9 downregulated pathways in hypoxic *Hif1α*$^{-/-}$ T cells found that all of them were either directly involved or intimately interacted with cellular metabolism. In contrast, none of these metabolic pathways were impacted in normoxic *Hif1α*$^{-/-}$ T cells, consistent with our previous finding of a non-essential role of HIF1α in metabolic reprogramming in normoxic T cells[8]. Among these top altered metabolic processes in hypoxic *Hif1α*$^{-/-}$ T cells (#3: central carbon metabolism, #4: fructose and mannose metabolism, #7: galactose metabolism, and #8: pentose phosphate pathway), glycolysis (#2) was the most heavily impacted metabolic pathway (Fig. 2C), with most significantly altered genes being in this pathway (e.g., *Slc2a3*, *Slc16a3*, *Tpi1*, *Slc2a1*, *Eno1*, *Pkm*, *Hk2*, *Ldha*, *Gpi1*, *Gapdh*, *Aldoc*, etc., Fig. S2E). These transcriptomic data pointed to an essential role of HIF1α in metabolic reprogramming of hypoxic T cells[8], especially glycolysis.

Next, we wanted to confirm the aforementioned transcriptomic alterations (RNA-seq data) by directly measuring mRNA expression of important glycolytic genes with real-time RT-PCR, which showed robust downregulation in hypoxic *Hif1α*$^{-/-}$ T cells (Fig. 2D). In keeping with the non-essential role of HIF1α in regulating glycolysis of normoxic T cells[8], their mRNA expression levels were modestly altered in normoxic *Hif1α*$^{-/-}$ T cells (Fig. 2D). We also assessed protein expression of select glycolytic molecules by Western blot (i.e., Glut1 – the major glucose transporter in T cells, hexokinase 2 (HK2) – a critical rate-limiting glycolytic enzyme catalyzing hexose phosphorylation, LDHa-the enzyme catalyzing the inter-conversion step of pyruvate to lactate in glycolysis, and phosphorylation of PDH (pyruvate dehydrogenase) (p-PDH)—a modification that inhibits the activity of PDH), which exhibited substantial alterations in hypoxic *Hif1α*$^{-/-}$ T cells but only modest or no changes in normoxic *Hif1α*$^{-/-}$ T cells (Fig. 2E). We also directly measured intracellular and extracellular metabolic flux using $^{13}$C-labeled glucose and found that *Hif1α*$^{-/-}$ T cells had significantly reduced glycolysis, with concomitant increase of oxidative phosphorylation (OxPhos) through the TCA cycle (Fig. 2F). This was associated with reduced uptake of glucose and secretion of lactate, and compensatory increases of uptake of extracellular glutamine and pyruvate (Supplementary Data 3). Along this line, hypoxic *Hif1α*$^{-/-}$

T cells had decreased extracellular acidification rate (ECAR, a readout for glycolysis) and increased oxygen consumption rate (OCR, a readout for OxPhos) by Seahorse analyzer (Fig. 2G). Taken together, these data establish HIF1α as a key regulator of metabolic reprogramming in hypoxic T cells, distinct from its dispensable role in normoxic T cells[8].

A recent study showed that aerobic glycolysis, under the control of LDHa[40], mediated IFN-γ induction in normoxic T cells. We therefore asked if anaerobic glycolysis of hypoxic T cells, under the regulation of HIF1α, controls IFN-γ production[6,7,40]. To test this, activated hypoxic T cells were treated with 2-DG, a well-established glycolytic inhibitor. As expected, 2-DG significantly suppressed glycolysis (ECAR) but not OxPhos (OCR) (Fig. 2G), as well as substantially reduced uptake of 2-NBDG (2-(N-(7-nitrobenz-2-oxa-1,3-diazol-4-yl)amino)−2-deoxyglucose), a fluorescent glucose analog (Fig. S2F). Remarkably, 2-DG almost completely abolished IFN-γ induction in both mouse (Fig. 2H) and human T cells (Fig. 2I), highlighting a crucial role of glycolysis in IFN-γ induction in hypoxic T cells. Similar to HIF1α deletion, 2-DG increased IL-2 production (Fig. S2G), mediating reciprocal regulation of late versus early cytokines in hypoxic T cells.

## Direct regulation of IFN-γ induction in hypoxic T cells by HIF1α and acetyl-CoA

Our above results support a crucial role of T cell HIF1α-glycolysis in IFN-γ induction under hypoxia, prompting us to determine a specific glycolytic control, downstream of HIF1α, that drives IFN-γ induction. We thus overexpressed individual key glycolytic molecules, i.e., Glut1, PKM2, LDHa, and MCT4, with a special focus on LDHa, because a reported role of LDHa in IFN-γ induction in normoxic T cells[40]. Unfortunately, overexpression of any of these individual molecules in hypoxic *Hif1α*$^{-/-}$ T cells (Fig. S3A) did not rescue IFN-γ reduction (Fig. S3B), suggesting this was unlikely mediated by a single glycolytic molecule but rather an outcome of a whole-spectrum suppression of the glycolytic pathway by HIF1α deletion. To establish a direct role of HIF1α in this process, we re-expressed HIF1α in *Hif1α*$^{-/-}$ T cells, using two complementary approaches: overexpression of either WT HIF1α or hydroxylation-defective triple mutant HIF1α (P402A/P577A/N813A) that is non-degradable and would remain constitutively active (HIF1α-TM)[41]. While retroviral transduction of T cells under hypoxia only led to modest increase of HIF1α expression, because of the stabilization of endogenous HIF1α, on the other hand, there was a markedly increased expression of HIF1α in *Hif1α*$^{-/-}$ T cells (Fig. 3A). Importantly, HIF1α re-expression largely re-stored IFN-γ production in *Hif1α*$^{-/-}$ T cells to that of WT T cells (Fig. 3B), supporting that HIF1α is necessary and sufficient to drive IFN-γ induction in hypoxic T cells.

A direct consequence of the significantly decreased glycolytic activity is the depletion of intracellular pool of acetyl-CoA ([acetyl-CoA])[40,42]. Inspired by a recent study showing that [acetyl-CoA] was instrumental for IFN-γ induction in normoxic *Ldha*$^{-/-}$ T cells[40], we asked if it was also responsible for impaired IFN-γ induction in hypoxic *Hif1α*$^{-/-}$ T cells. Indeed, hypoxic *Hif1α*$^{-/-}$ T cells had markedly reduced [acetyl-CoA] (Fig. 3C). Since acetyl-CoA can be regenerated from acetate by acetyl-CoA synthetase in T cells[40], independent of citrate release from mitochondria, we added sodium acetate (NaAc) to the cultures of activated WT and *Hif1α*$^{-/-}$ T cells on Day 2, prior to appreciable IFN-γ induction on Day 3. Cells were harvested on Day 5 to measure [acetyl-CoA] and detect IFN-γ production. Not only did NaAc replenish [acetyl-CoA] in *Hif1α*$^{-/-}$ T cells to the level of WT T cells (Fig. 3D), but more importantly, this fully restored IFN-γ production in *Hif1α*$^{-/-}$ T cells (Fig. 3E), supporting that the maintenance of [acetyl-CoA] in hypoxic T cells by HIF1α-glycolysis axis was a major metabolic mechanism underscoring IFN-γ induction. Considering the essential role of IFN-γ in anti-tumor responses[43-45], we evaluated the ability of *Hif1α*$^{-/-}$ T cells to kill tumor cells by co-culturing pre-activate WT or *Hif1α*$^{-/-}$ hypoxic T cell with MB49 bladder tumor cells at a ratio of 2:1. Cell death of MB49

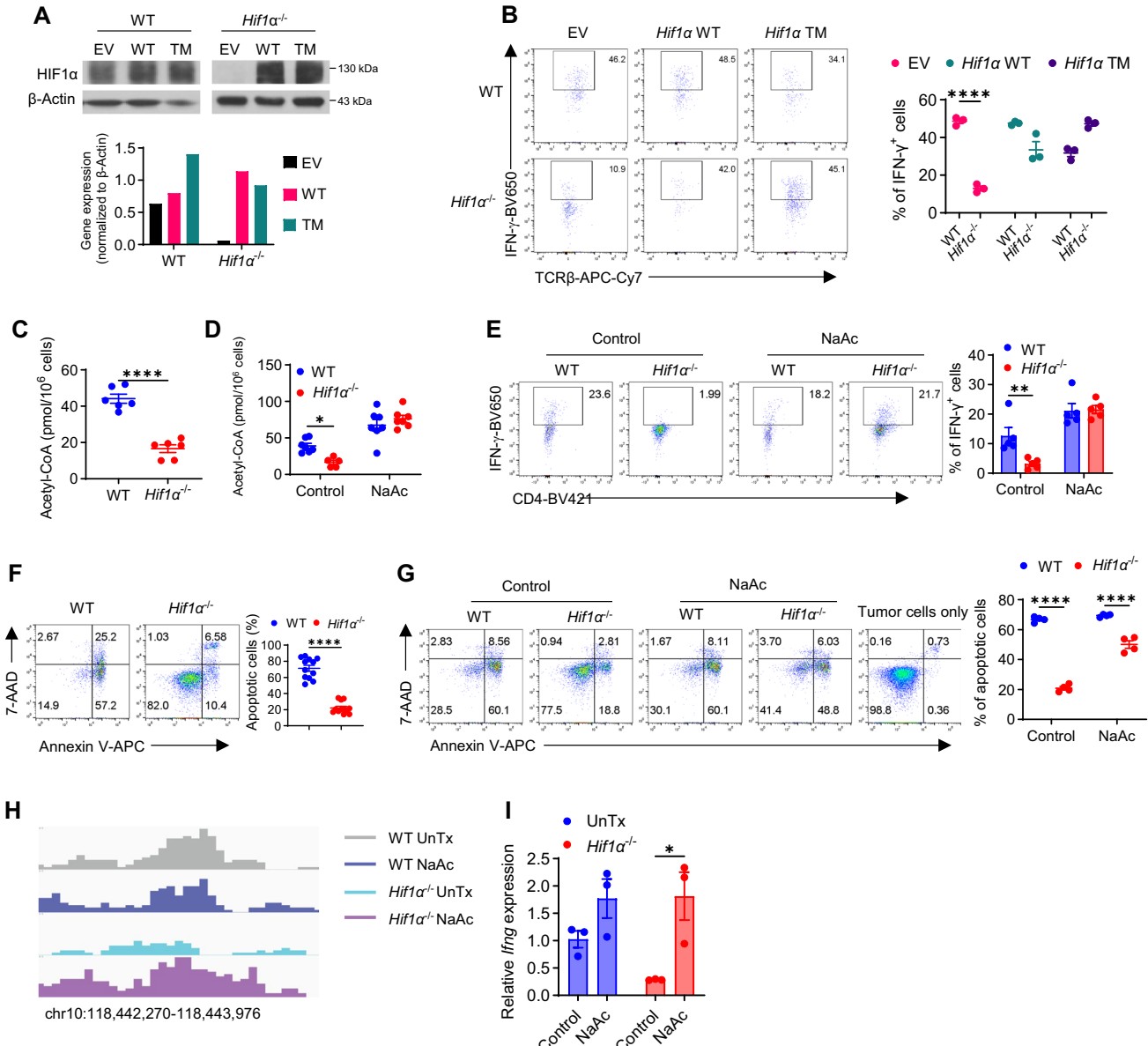

**Fig. 3 | Restoration of HIF1α and intracellular acetyl-CoA rescues IFN-γ production in *Hif1α⁻/⁻* T cells. A, B** Protein expression of HIF1α in WT and *Hif1α⁻/⁻* CD4⁺ T cells successfully transduced (GFP⁺) with empty retroviruses (EV) or retroviruses expressing WT or triple-mutant *Hif1α* (TM) under hypoxia (**A**). GFP⁺ T cells from (**A**) were activated under hypoxia and analyzed for IFN-γ production (**B**) (*N* = 3 per group, ****p < 0.0001)). **C** Intracellular acetyl-CoA in activated WT and *Hif1α⁻/⁻* CD4⁺ T cells (*N* = 6 per group, ****p < 0.0001). **D–E** Intracellular acetyl-CoA (**D**) (*N* = 6 per group, *p = 0.022) and IFN-γ production (**E**) (*N* = 5 per group, **p = 0.0081) by activated WT and *Hif1α⁻/⁻* CD4⁺ T cells, cultured with or without 20 mM sodium acetate (NaAc) added on Day 2 post-activation. **F** Cell death of MB49 cells cultured with activated WT and *Hif1α⁻/⁻* CD4⁺ T cells at the ratio of 1:2 for 48 h was analyzed by 7-AAD/Annexin V staining (*N* = 12 per group, ****p < 0.0001). **G** Cell death of

MB49 cells co-cultured with activated WT and *Hif1α⁻/⁻* CD4⁺ T cells pre-treated with or without NaAc for 48 h was analyzed by 7-AAD/Annexin V staining (*N* = 4 per group, ****p < 0.0001). **H,I** H3K9Ac enrichment at *Ifng* promoter and upstream regions (**H**) and *Ifng* mRNA expression (**I**) (*N* = 3 per group, *p = 0.0123) in activated WT and *Hif1α⁻/⁻* CD4⁺ T cells treated with solvent (UnTx) or 20 mM of NaAc. A two-sided Student's t-test was used in (**C** and **F**) for statistical analyzes. Two-way ANOVA with Šídák's multiple comparisons test (with adjustment) was used for (**B, D–E, G**) and **I**. All experiments were repeated at least twice. Pooled results shown in the dot plots and bar graphs depicted means ± SEM for all the samples in each group, with each dot denoting an independent sample. Source data were provided in the Source Data file.

cells was analyzed 48 h later by staining for Annexin V and 7-AAD, two commonly used markers for early and late apoptotic/necrotic cells. In line with reduced IFN-γ production (Fig. S3C) in *Hif1α⁻/⁻* T cells, they were less able to kill tumor cells (Fig. 3F) than WT T cells. Moreover, in compliance with restored IFN-γ production in *Hif1α⁻/⁻* T cells treated with NaAc, their tumor-killing capacity was restored as well (Fig. 3G).

To understand how acetate supplementation rescued IFN-γ production in *Hif1α⁻/⁻* T cells, we first measured ECAR and OCR in hypoxic

WT and *Hif1α⁻/⁻* T cells, treated with or without NaAc. As shown in Fig. S3D, NaAc did not bolster glycolysis in *Hif1α⁻/⁻* T cells, consistent with previous reports showing that depletion of [acetyl-CoA] was a downstream consequence of decreased glycolytic activity[40,42]. Interestingly, increased [acetyl-CoA] upon acetate supplementation acted as a negative feedback mechanism, resulting in inhibited glycolysis and OxPhos in both WT and *Hif1α⁻/⁻* T cells. This was further supported by reduced 2-NBDG uptake (Fig. S3E) and glucose consumption (Fig. S3F)

by NaAc. Thus, acetate supplementation did not seem to rescue IFN-γ production in *Hif1α*[−/−] T cells by boosting glycolysis. Next, considering that histone acetylation is an important epigenetic mechanism regulating gene expression[46], including *Ifng*, dependent on [acetyl-CoA][40], we asked whether NaAc, by restoring [acetyl-CoA] in *Hif1α*[−/−] T cells, induced histone acetylation and upregulated *Ifng* expression. To this end, we performed CUT&RUN analysis of histone H3 acetylation at the lysine 9 residue (H3K9Ac), a histone mark associated with active transcribed genes in normoxic T$_H$1 cells[40]. As expected, in line with drastically reduced IFN-γ, hypoxic *Hif1α*[−/−] T cells had significantly diminished H3K9Ac in the promoter and upstream regulatory regions of *Ifng* gene. Acetate supplementation greatly increased H3K9Ac in hypoxic *Hif1α*[−/−] T cells (Fig. 3H), coupled with significantly increased transcription of *Ifng* gene (Fig. 3I). Consistent with no overt impact of acetate supplementation on glycolysis in *Hif1α*[−/−] T cells, NaAc did not change H3K9Ac in the loci of *Glut1* and *Ldha* genes, which exhibited expected reduction in *Hif1α*[−/−] T cells (Fig. S3G), reflecting their inactive transcription in the absence of HIF1α. All the regions with differentially-expressed H3K9Ac in WT and *Hif1α*[−/−] T cells treated with or without NaAc were provided in Supplementary Data 4.

In addition to acetate supplementation, a recent study found that ketone bodies like β-hydroxybutyrate (BHB) can serve as a substrate for acetyl-CoA production in effector T cells[47]. BHB was impaired in CoVID-19 patients but not influenza patients with acute respiratory stress (ARS), and BHB supplementation increased IFN-γ production in T$_H$1 and T$_C$1 cells, under normoxia. To assess whether BHB supplementation, like acetate supplementation, rescued the impaired IFN-γ induction in hypoxic *Hif1α*[−/−] T cells, cells were treated with 1 and 5 mM BHB but this did not rescue the reduced IFN-γ production in hypoxic *Hif1α*[−/−] T cells Fig. S3H, suggesting a selective rescuing effect of acetate supplementation. Collectively, we show that HIF1α, by maintaining glycolysis and [acetyl-CoA], controls effector function and tumor-killing ability of hypoxic T cells. Acetate supplementation, by restoring [acetyl CoA] and H3K9Ac expression in the promoter and regulatory regions of *Ifng* gene in *Hif1α*[−/−] T cells, enhances *Ifng* transcription and rescues IFN-γ production as well as tumor-killing capacity of *Hif1α*[−/−] T cells.

## Impaired IFN-γ induction in *Hif1α*[−/−] T cells is not due to their proliferative defect

Next, we wanted to define cellular mechanisms underpinning the impaired IFN-γ induction in hypoxic *Hif1α*[−/−] T cells. Considering that glycolysis is an essential component of metabolic reprogramming during T cell activation[8] and the drastically reduced glycolysis in hypoxic *Hif1α*[−/−] T cells, we posited that their activation would be substantially impaired. To this end, we measured two widely used markers for T cell activation: inducible T-cell COstimulator (ICOS) and CD25. In agreement with a largely dispensable role of HIF1α in the metabolic reprogramming of normoxic T cell[8], no overt changes to ICOS and CD25 expression in normoxic *Hif1α*[−/−] T cells were observed; in stark contrast, their expression was markedly reduced in hypoxic *Hif1α*[−/−] T cells (Fig. 4A). Another cardinal feature of less active T cells is their smaller cell size[48], which can be measured by forward scatter (FSC). Again, only hypoxic but not normoxic *Hif1α*[−/−] T cells appeared smaller than their WT counterparts (Fig. 4B). Glycolysis^low (Glut1^low) T cells had significantly reduced CD25 and ICOS expression than glycolysis^high (Glut1^high) cells in both CD4^+ (Fig. S4A) and CD8^+ (Fig. S4B) T cells, directly correlating glycolysis with T cell activation[8]. Since CD25 is a marker for T$_{reg}$, we stained cells for FoxP3 and observed comparable frequencies of FoxP3^+ T$_{reg}$ between WT and *Hif1α*[−/−] T cells (Fig. S4C), ruling out the possibility that reduced CD25 expression in *Hif1α*[−/−] T cells was due to the differential abundance of T$_{reg}$. These results together showed that *Hif1α*[−/−] hypoxic T cells, with decreased glycolysis, had impaired T cell activation.

T cell activation initiates a series of intracellular events that drive cell proliferation. Indeed, less active *Hif1α*[−/−] T cells proliferated slower, evidenced by decreased Ki-67 staining (Fig. S4D). To further assess this, WT and *Hif1α*[−/−] naïve CD4^+ T cells were labeled with CellTrace Violet (CTV), whose dilution distinctively demarcates cell divisions. Consistent with normal activation of normoxic *Hif1α*[−/−] T cells, there was no overt alteration of cell proliferation. On the other hand, less active hypoxic *Hif1α*[−/−] T cells exhibited substantially reduced proliferation. The proliferative defect in *Hif1α*[−/−] T cells was evident as early as Day 2 (Fig. 4C), shortly after the initial growth phase in activated T cells (~24 h)[8], which was also observed under a different hypoxic condition (2.5% O$_2$, Fig. S4E). Acetate supplementation did not rescue the proliferative defect (Fig. S4F), suggesting reduced [acetyl CoA] was a downstream outcome of glycolysis-driven cell activation and proliferation. Similar proliferative defect was observed in hypoxic *Hif1α*[−/−] CD8^+ T cells (Fig. S4G). Given the reported intimate relationship of T cell proliferation to IFN-γ production[49], we asked if the proliferative defect in hypoxic *Hif1α*[−/−] T cells underpinned IFN-γ reduction. As shown in Fig. 4D, regardless of cell division, *Hif1α*[−/−] T cells produced IFN-γ at a much reduced level compared to WT T cells. In fact, *Hif1α*[−/−] T cells, even with higher divisions, still produced substantially less IFN-γ than WT T cells with lower divisions (the bar graph on the right), strongly arguing that impaired IFN-γ production was not due to defective proliferation of *Hif1α*[−/−] T cells.

## HIF1α-glycolysis-driven AICD controls IFN-γ induction in hypoxic T cells

Another major outcome following T cell activation is AICD, which is commonly regarded as a housekeeping process to remove obsolete effector T cells during the contraction phase, following a successful immune response. When gone awry, this could disrupt the immune homeostasis, causing autoimmune diseases[50] and/or breaching transplantation tolerance[51]. But whether AICD governs the effector function of T cells (e.g., IFN-γ induction) is unknown. Given the weaker activation of hypoxic *Hif1α*[−/−] T cells, we contemplated that their AICD would be attenuated. Following T cell activation under hypoxia (1% O$_2$), cell death of WT and *Hif1α*[−/−] T cells were detected at different times. While modest cell death was observed on Day 1-2, it rapidly arose on Day 3 (approximately, 56 h post-activation) with few live WT T cells remaining (~9%, Fig. S5A), but this process was considerably delayed in the absence of HIF1α, as greater than 40% of *Hif1α*[−/−] T cells stayed alive on Day 3, with an increasing trend on Day 4 and Day 5. Similar observation was made at 2.5% O$_2$ (Fig. S5B).

To understand why *Hif1α*[−/−] T cells survived better, considering that *Hif1α*[−/−] T cells were less metabolically active and would consume less nutrients in the medium, we wondered if this was simply due to greater availability of nutrients. To test this idea, we replaced half of the old media with fresh nutritious media daily, which did not change the AICD difference between WT and in *Hif1α*[−/−] T cells (Fig. S5C). Since *Hif1α*[−/−] T cells produced more IL-2 (Fig. S1G), a T cell growth factor[51], we therefore blocked IL-2 with neutralizing anti-IL2 antibodies that we previously used[52], which did not impact AICD in *Hif1α*[−/−] T cells either (Fig. S5D). Moreover, knowing that IFN-γ can induce AICD[53] and hypoxic *Hif1α*[−/−] T cells produced less amount of IFN-γ, we added recombinant IFN-γ to the cultures of hypoxic *Hif1α*[−/−] T cell to evaluate if this would rectify impaired AICD, but IFN-γ supplementation did not drive AICD in *Hif1α*[−/−] T cells, even at high doses (Fig. S5E). Collectively, these results indicated that inhibited AICD in hypoxic *Hif1α*[−/−] T cells was unlikely mediated by extrinsic factors but rather a direct outcome of attenuated glycolytic activity in these cells. In support of this conception, glycolytic inhibition with 2-DG suppressed AICD in both mouse (Fig. 5A) and human hypoxic T cells (Fig. S5F).

Next, we asked if impaired AICD in *Hif1α*[−/−] hypoxic T cells could contribute to IFN-γ reduction. Along this line, we observed an intimate correlation of impaired AICD and reduced IFN-γ

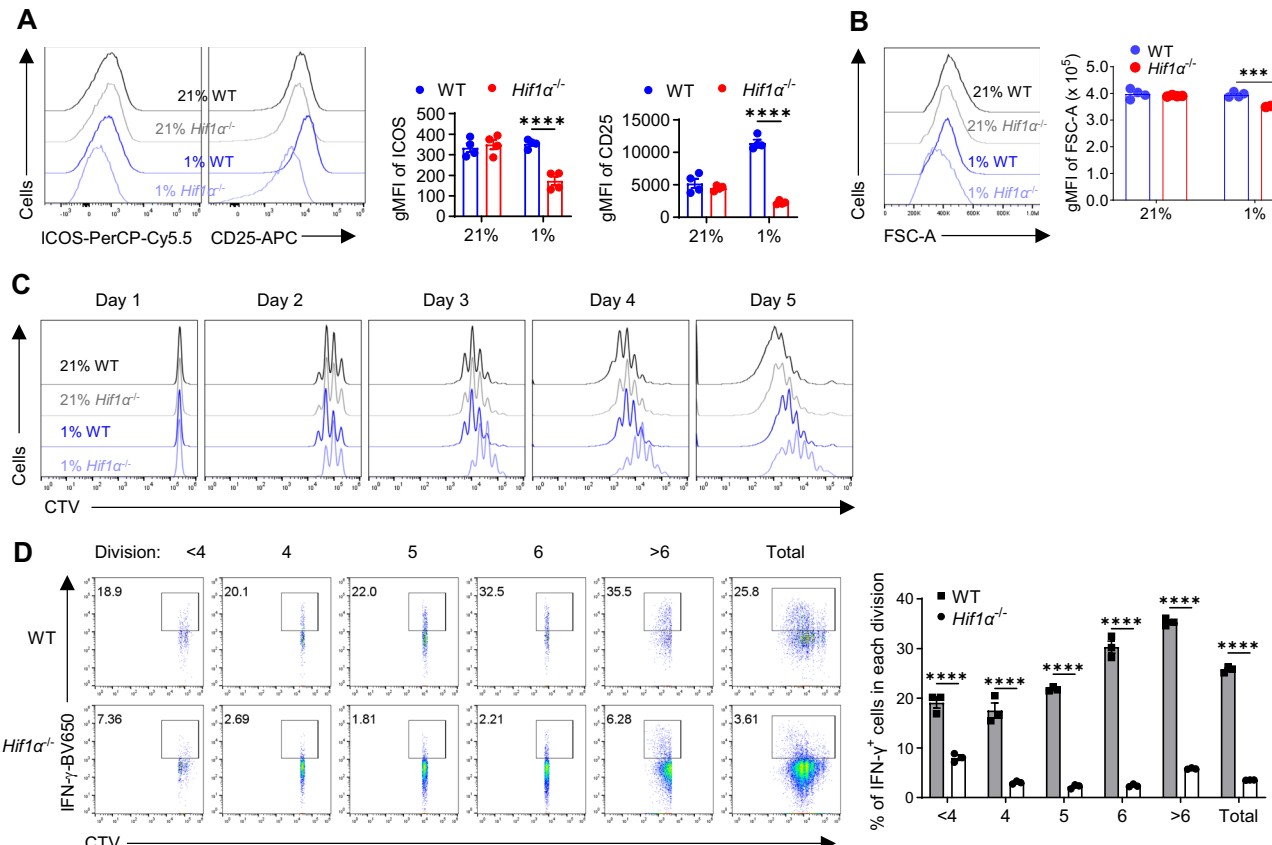

**Fig. 4 | Reduced IFN-γ production in *Hif1α*⁻/⁻ T cells is not due to proliferative defect. A**, **B** Naïve WT and *Hif1α*⁻/⁻ CD4⁺ T cells were activated under normoxia (21% $O_2$) and hypoxia (1% $O_2$) for 5 days. Gated live cells were analyzed for the expression of ICOS and CD25 (**A**), depicted as geometric mean fluorescence intensity (gMFI), and area of forward scatter (FSC-A) (**B**) (*N* = 4 per group, ****$p < 0.0001$, ***$p = 0.0001$). **C**, **D** Naïve WT and *Hif1α*⁻/⁻ CD4⁺ T cells were labeled with CellTrace Violet (CTV) and activated under normoxia (21% $O_2$) and hypoxia (1% $O_2$). CTV dilution was monitored daily to assess cell proliferation (**C**). IFN-γ production by activated WT and *Hif1α*⁻/⁻ CD4⁺ T cells within indicated cell divisions was shown (**D**) (*N* = 3 per group, ****$p < 0.0001$). Two-way ANOVA with Šídák's multiple comparisons test (with adjustment) was used for (**A**, **B** and **D**). All the experiments were repeated at least twice. Pooled results shown in the dot plots and bar graphs depicted means ± SEM for all the samples in each group, with each dot denoting an independent sample. Source data were provided in the Source Data file.

induction, which coincidentally occurred in hypoxic but not in normoxic *Hif1α*⁻/⁻ T cells (Fig. 5B). Inspired by this, we directly tested the role of AICD in IFN-γ induction, by blocking AICD with z-VAD-fmk (Fig. 5C), a cell-permeable, irreversible pan-caspase inhibitor[51]. Remarkably, this also drastically blocked IFN-γ production (Fig. 5D). Even with the low level of IFN-γ production in *Hif1α*⁻/⁻ T cells, z-VAD-fmk was able to further block it (Figs. 5D, 1.34% versus 0.26% IFN-γ⁺ cells), in association with a modest yet significant suppression of AICD (Figs. 5C, 53.5% versus 73.8% live cells). Lastly, because NaAc rescued IFN-γ induction in hypoxic *Hif1α*⁻/⁻ T cells, we wondered if NaAc could activate AICD in hypoxic *Hif1α*⁻/⁻ T cells, which appeared to be the case (Fig. 5E). More importantly, NaAc-induced AICD led to significantly increased IFN-γ production in *Hif1α*⁻/⁻ T cells (Fig. 5F). Collectively, these data establish that the HIF1α-glycolysis-acetyl CoA axis engages AICD in hypoxic T cells to drive IFN-γ induction.

**Loss of HIF1α selectively impaired IFN-γ production in TILs**

Having demonstrated a selective role of HIF1α in IFN-γ induction in hypoxic T cells in vitro, we wanted to recapitulate this in vivo. As known, TME in solid tumors is a highly hypoxic milieu, with $O_2$ levels being ~1%[27]; on the other hand, peripheral lymphoid organs, like spleens and draining lymph nodes (DLN), represent more oxygenated environments[22,23]. We reason that peripheral T cells from spleens/DLNs and TILs form a natural, albeit not ideal in vivo system. To this end, WT and *Hif1α*⁻/⁻ mice were inoculated with MB49 bladder tumor cells. Once established tumors formed, TILs and peripheral T cells were subject to cytokine analyzes. Whereas there was no overt reduction of IFN-γ in *Hif1α*⁻/⁻ peripheral T cells from spleen (Fig. 6A) and DLN (Fig. S6A), it was considerably lower in *Hif1α*⁻/⁻ TILs (Fig. 6B), in comparison to their WT counterparts, corroborating our in vitro data showing a selective role of HIF1α in hypoxic T cells. Likewise, GzmB (Fig. S6B) and Prf (Fig. S6C) were significantly reduced in *Hif1α*⁻/⁻ TILs but not peripheral T cells. Moreover, similar to greater production of IFN-γ, GzmB, and Prf by hypoxic T cells than normoxic T cells in vitro, TILs produced more cytokines than peripheral T cells (Fig. S6D). Comparable results were obtained in the orthotopic B16-BL6 melanoma model (Fig. 6D, C).

*Vhl*⁻/⁻ T cells with stabilized HIF1α were predisposed to produce more IFN-γ as early as Day 2, when activated in vitro (Fig. 1K). To assess this in vivo, we inoculated WT and *Vhl*⁻/⁻ mice with MB49 tumor cells. TILs and peripheral T cells were similarly analyzed, as described above. As expected, *Vhl*⁻/⁻ T cells (both TILs and peripheral T cells) had increased expression of Glut1 (Fig. S6E)[7], as a result HIF1α stabilization and increased function. Correspondingly, both *Vhl*⁻/⁻ TILs (Fig. 6F) and peripheral T cells from spleen (Fig. 6E) as well as DLNs (Fig. S6F) had increased IFN-γ production than their WT counterparts. Altogether, these in vivo results corroborated our in vitro data, establishing a pivotal role of HIF1α in controlling IFN-γ production in hypoxic T cells.

**HIF1α in T cells governs ICB efficacy**

ICBs have emerged as a major pillar of cancer care[54–59]. While it is recognized that functional rejuvenation (e.g., IFN-γ production) of TILs

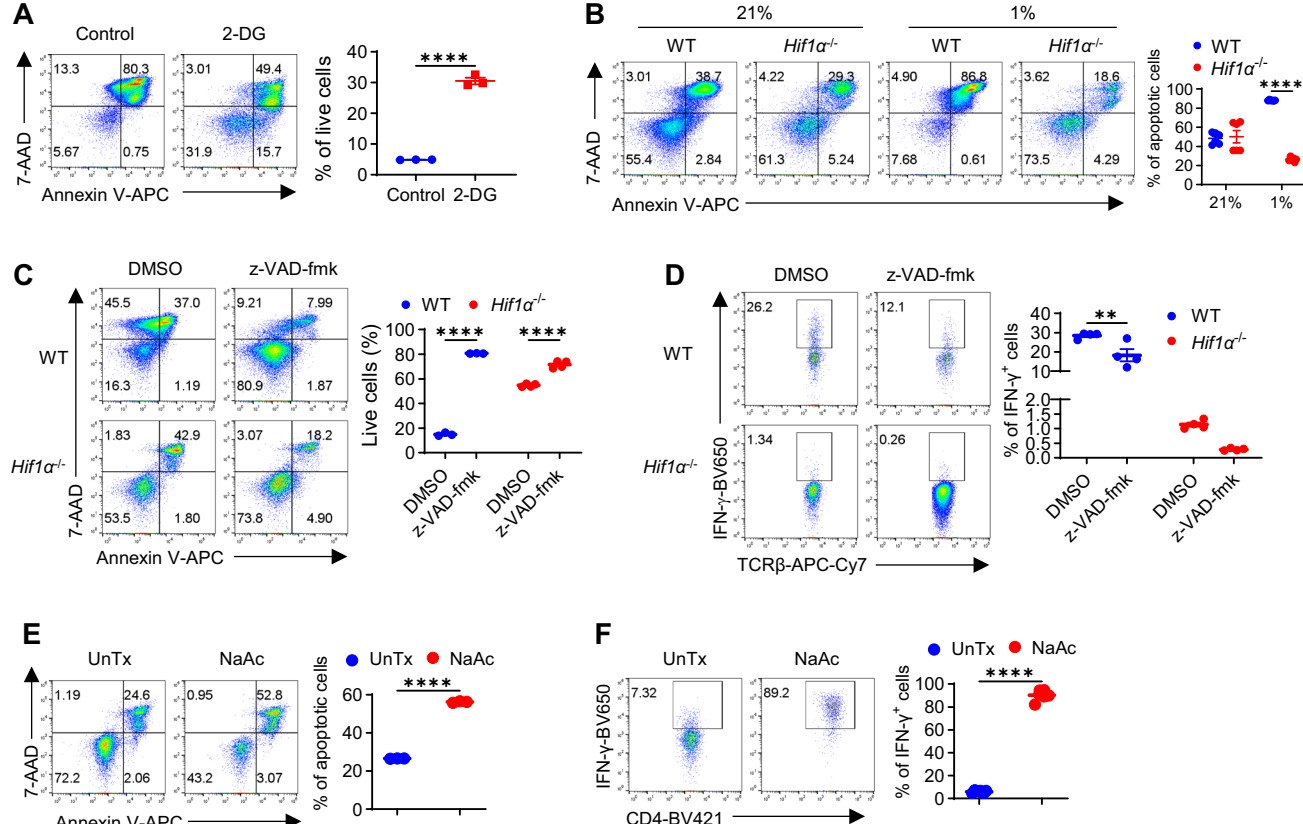

**Fig. 5 | HIF1α-glycolysis-driven AICD governs IFN-γ production in hypoxic T cells. A** Cell death of naïve WT CD4⁺ T cells activated under hypoxia for 3 days, with or without with 2-DG was measured by 7-AAD and Annexin V staining ($N = 3$ per group, ****$p < 0.0001$). **B** Cell death of naïve WT and $Hif1\alpha^{-/-}$ CD4⁺ T cells activated under normoxia (21% O₂) and hypoxia (1% O₂) for 3 days was detected by 7-AAD/ Annexin V staining ($N = 6$ per group, ****$p < 0.0001$). **C, D** Naïve WT and $Hif1\alpha^{-/-}$ CD4⁺ T cells were activated under hypoxia, with z-VAD-fmk or without (DMSO); on day 3, cells were stained for 7-AAD and Annexin V to assess cell death (**C**) ($N = 4$ per group, ****$p < 0.0001$), and on Day 5, IFN-γ production was determined (**D**) ($N = 4$ per group, **$p = 0.0016$). **E, F** Naïve $Hif1\alpha^{-/-}$ CD4⁺ T cells were activated under hypoxia, with or without 20 mM sodium acetate (NaAc) added on day 0; on Day 3, cells were stained for 7-AAD and Annexin V to assess cell death (**E**) ($N = 3$ per group, ****$p < 0.0001$), and IFN-γ production was determined on Day 5 (**F**) ($N = 6$ per group, ****$p < 0.0001$). A two-sided Student's t-test was used in (**A**, **E** and **F**) for statistical analyzes. Two-way ANOVA with Šídák's multiple comparisons test (with adjustment) was used for (**B**–**D**). All the experiments were repeated at least twice. Pooled results shown in the dot plots depicted means ± SEM for all the samples in each group, with each dot denoting an independent sample. Source data were provided in the Source Data file.

by ICBs governs their efficacy[43,45,60], specific T cell-intrinsic mechanisms have not been well understood. Our results showed that HIF1α in T cells controlled the effector functions of hypoxic T cells, including TILs. We contemplated that T cell HIF1α would represent an essential metabolic and molecular mechanism underlying ICBs. To test this, we treated WT and $Hif1\alpha^{-/-}$ mice bearing palpable MB49 bladder tumors with combined anti-CTLA-4+anti-PD-1 (combo ICB), a more efficacious ICB therapy than monotherapies (anti-CTLA-4 or anti-PD-1 alone)[61], following the regimen that we previously described[43]. As shown in Fig. 7A, while combo ICB potently suppressed tumor growth in WT mice, it failed to do so in $Hif1\alpha^{-/-}$ mice, revealing T cell-intrinsic HIF1α pathway as a major functional mechanism in ICBs. Tumors from combo ICB-treated WT mice were visually smaller and weighed significantly less than those from $Hif1\alpha^{-/-}$ mice (Fig. 7B). When isolated T cells from these mice were analyzed for IFN-γ production, it was greatly reduced in $Hif1\alpha^{-/-}$ TILs (Fig. 7C) but not in peripheral $Hif1\alpha^{-/-}$ T cells from spleen (Fig. S7A) and DLNs (Fig. S7B) compared to their WT counterparts, again confirming a selective role of HIF1α in driving IFN-γ production in hypoxic T cells. Our results were consistent with an early study showing that HIF1α in CD8⁺ T cells was required for optimal anti-tumor immunity in colorectal cancer (MC38)[62], following adoptive transfer of CD8⁺ T cells and ICBs. Together, these studies establish a pivotal role of T cell-intrinsic HIF1α signaling in governing anti-tumor immunity elicited by ICBs in different types of tumors.

Next, considering that glycolytic inhibition by 2-DG almost completely abolished IFN-γ production in T cells (Fig. 2H), we postulated that it would abrogate therapeutic effects of ICBs in tumor-bearing mice. To this end, WT and $Hif1\alpha^{-/-}$ tumor-bearing mice treated with combo ICB were fed with 2-DG containing drinking water, *ad libitum*. Tumor growth was closely monitored. Clearly, in vivo administration of 2-DG greatly diminished tumor suppression by combo ICB in tumor-bearing WT mice (Fig. 7D), together with reduced IFN-γ production (Fig. S7C). 2-DG effects were weaker in $Hif1\alpha^{-/-}$ mice (Figs. 7D and S7C), highlighting expected dependence of 2-DG effects on T cell HIF1α pathway. Together, these data support a pivotal role of T cell HIF1α-glycolysis axis in dictating ICB efficacy.

A significant barrier in ICBs is therapeutic resistance[61,63]. Our above results, together with a previous study[62], identify loss of the HIF1α-glycolysis axis in T cells as a major T cell-intrinsic mechanism of ICB resistance. However, how to overcome this ICB resistance has not been explored, a solution to which would undoubtedly improve the utility of ICBs so that more patients can benefit from these novel immunotherapeutics. Inspired by previous studies[64,65] and our data showing that acetate supplementation could rescue IFN-γ production and tumor killing ability of $Hif1\alpha^{-/-}$ T cells in vitro, we asked if this could act as an effective strategy to bypass ICB resistance in $Hif1\alpha^{-/-}$ tumor-bearing mice in vivo. To test this, tumor-bearing WT and $Hif1\alpha^{-/-}$ mice were treated with acetate supplementation, followed by

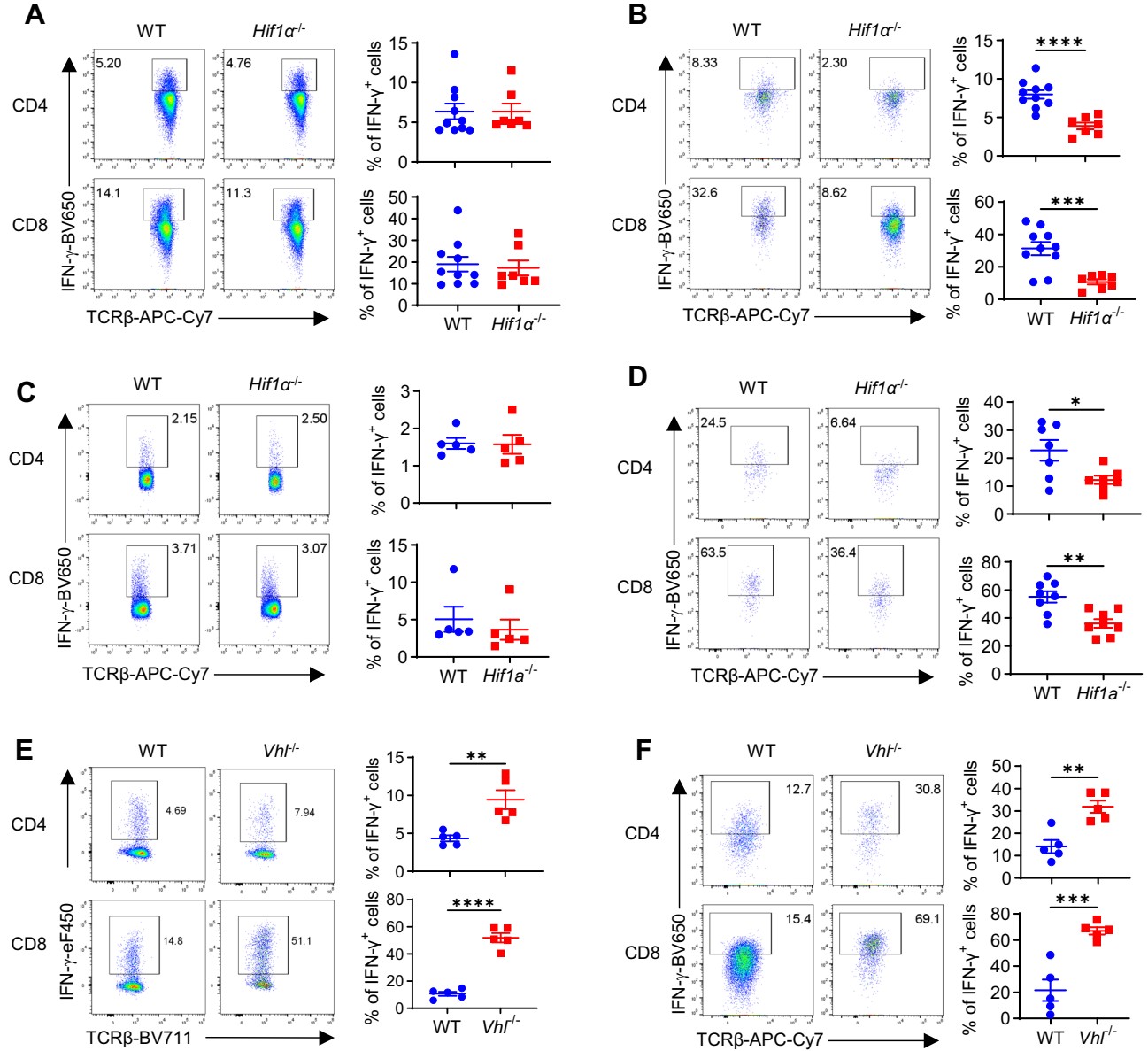

**Fig. 6 | HIF1α regulates IFN-γ production in tumor-infiltrating T cells (TILs).**
**A**, **B** T cells isolated from WT ($N = 10$) or $Hif1α^{-/-}$ ($N = 7$) mice bearing established MB49 bladder tumor were analyzed for IFN-γ production by CD4+ and CD8+ splenocytes (**A**) or TILs (**B**) (**p < 0.0001, *p = 0.001). **C**, **D** T cells isolated from WT or $Hif1α^{-/-}$ mice bearing established orthotopic B16-BL6 melanoma were analyzed for IFN-γ production by CD4+ and CD8+ splenocytes (**C**) ($N = 5$ per group) or TILs (**D**) ($N = 7$ per group, *p = 0.0042, **p < 0.0024). **E**, **F** T cells isolated from WT or

$Vhl^{-/-}$ mice bearing established MB49 bladder tumor were analyzed for IFN-γ production by CD4+ and CD8+ splenocytes (**E**) ($N = 5$ per group, **p = 0.0097, ****p < 0.0001) or TILs (**F**) ($N = 5$ per group, **p = 0.0021, ***p = 0.0008). A two-sided Student's t-test was used in (**A**–**F**) for statistical analyzes. All the experiments were repeated 2–5 times. Pooled results shown in the dot plots depicted means ± SEM for all the mice in each group, with each dot denoting an individual mouse. Source data were provided in the Source Data file.

combo ICB therapy. While acetate supplementation did not further improve ICB efficacy in WT tumor-bearing mice, it significantly improved therapeutic effects of ICB in $Hif1α^{-/-}$ tumor-bearing mice, evidenced by potent suppression of tumor growth (Fig. 7E), greatly reduced tumor weights (Fig. 7F), and significantly increased IFN-γ production in $Hif1α^{-/-}$ CD4+ (Fig. 7G) and CD8+ (Fig. S7D) TILs. In summary, loss of HIF1α in T cells impairs their tumor killing capacity and renders tumor-bearing mice resistant to ICBs, revealing a major T cell-intrinsic mechanism of therapeutic resistance to ICBs, which can be overcome by acetate supplementation (Fig. S7E).

## Discussion

Metabolic reprogramming emerges as the fundamental driving force in T cell activation and differentiation[5–15,17], of which glycolysis

is a key component. Concerted efforts from others and us show that glycolysis is essential for IFN-γ induction in T cells under normoxia[6,7,40], but this is independent of HIF1α[7,19]. It remains to be determined whether and how HIF1α regulates IFN-γ induction in T cells under hypoxia, a condition commonly found in various pathophysiological settings, such as TME in solid tumors, the lumen of the GI tract, and inflammatory sites[33]. Combining genetic as well as pharmacological approaches, we show here that HIF1α is indispensable for IFN-γ induction in naïve T cells activated under hypoxia, regulated by AICD but not activation-induced cell proliferation. Furthermore, HIF1α controls metabolic reprogramming in hypoxic T cells, distinct from its dispensable role in normoxic T cells. By sustaining glycolysis, HIF1α maintains [acetyl-CoA], IFN-γ induction, tumor-killing ability of hypoxic T cells, and tumor responses to

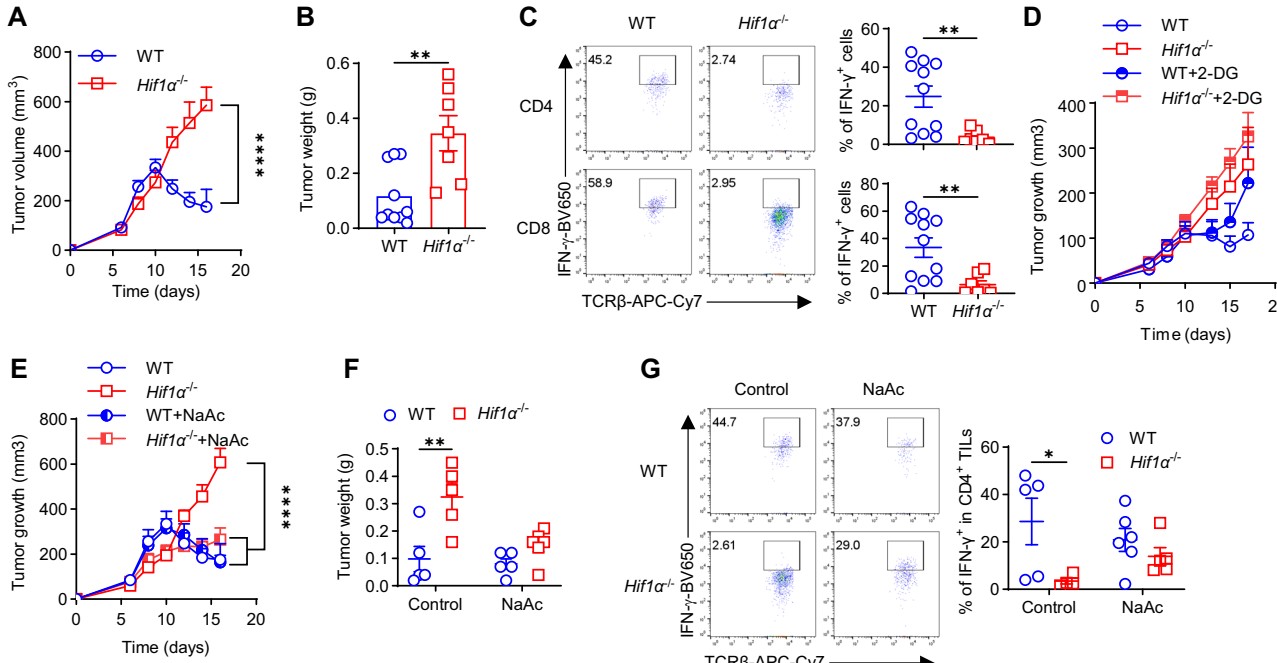

**Fig. 7 | HIF1α in T cells governs therapeutic effects of ICB. A, B** WT and *Hif1α⁻/⁻* mice bearing palpable MB49 bladder tumor were treated with combined anti-CTLA-4+anti-PD-1 therapy (ICB), followed by periodic measurement of tumor volume (**A**) and tumor weights at euthanization (**B**). **C** IFN-γ production by CD4⁺ and CD8⁺ TILs isolated from tumor-bearing mice in (**A–B**) (*N* = 10 for WT, *N* = 7 for *Hif1α⁻/⁻* (**A–C**), ****p < 0.0001 for (**A**), **p = 0.0037 for (**B**), **p = 0.0083 for CD4⁺ TILs and **p = 0.0097 for CD8⁺ TILs (**C**). **D** MB49 bladder tumor-bearing WT and *Hif1α⁻/⁻* mice treated with ICB alone or in conjunction with 2-DG administration, followed by periodic measurement of tumor volume. **E–F** WT and *Hif1α⁻/⁻* mice bearing palpable MB49 bladder tumor were treated with ICB alone or in

conjunction with administration of sodium acetate (NaAc), followed by periodic measurement of tumor volume (**E**) and tumor weights at euthanization (**F**) (*N* = 5 per group for (**E** and **F**), ****p < 0.0001, **p = 0.0041) (**G**). IFN-γ production by CD4⁺ TILs from tumor-bearing mice in (**E–F**) (*N* = 5 per group, *p = 0.0265). A two-sided Student's t-test was used in (**B, C**) for statistical analyzes. Two-way ANOVA with Šídák's multiple comparisons test (with adjustment) was used for (**A, D–G**). All the experiments were repeated 2–5 times. Pooled results shown in the dot plots depicted means ± SEM for all the mice in each group, with each dot denoting an individual mouse. Source data were provided in the Source Data file.

ICB, identifying loss of HIF1α as a major T cell-intrinsic mechanism of ICB resistance, which can be overcome by acetate supplementation.

To the best of our knowledge, this is the first study to show that HIF1α is a bona fide metabolic regulator of IFN-γ induction in hypoxic T cells. It also mediates the production of other late effector cytokines (e.g., Prf and GzmB) but negatively regulates production of early cytokines (i.e., IL-2 and TNF[38,39]), pointing to a reciprocal regulatory role of HIF1α in the production of late versus early cytokines in hypoxic T cells, which warrants future investigations to delineate the specific underlying mechanisms. Our finding of HIF1α as positive regulator of IFN-γ induction in hypoxic T cells is in line with previous reports showing that specific deletion of VHL in T_reg cells[41] increases IFN-γ production, VHL deletion[66] or PHD deletion[67] in T cells promotes the polyfunctionality of CD8⁺ T cells (i.e., increased production of IFN-γ, GzmB, etc.), and total (not naïve) CD8⁺ T cells activated under hypoxia produce more IFN-γ[68]. In contrast, two other studies[34,35] reported a negative regulation of IFN-γ production in CD4⁺CD25⁻ T cells by hypoxia and HIF1α, which may be because ~40–50% of CD4⁺CD25⁻ cells are already activated effector/central memory T cells (CD44⁺)[36,37] and equipped to produce IFN-γ. Nevertheless, our results support a key role of HIF1α in IFN-γ induction in naïve T cells activated under hypoxia.

An early study[40] elegantly demonstrates that LDHa but not HIF1α controls glycolysis and IFN-γ induction in normoxic T cells. On the other hand, we show that HIF1α but not LDHa drives glycolysis and IFN-γ induction in hypoxic T cells, highlighting distinctive metabolic controls of IFN-γ induction in normoxic versus hypoxic T cells, although they converge on glycolysis. Glycolysis is an essential component of metabolic reprogramming during T cell activation. In normoxic T cells, this is primarily mediated by Myc but not HIF1α[8], as deletion of HIF1α

doesn't affect glycolysis, cell growth, cell activation, and activation-driven cell proliferation, as we previously demonstrated[8] and in this study. However, in hypoxic T cells, HIF1α is a key regulator of activation-induced metabolic reprogramming. It would be interesting to investigate whether Myc also plays a role in hypoxic T cells and if so, how HIF1α interacts with Myc to achieve a delicate control of this complicated reprogramming.

Although attenuated glycolysis in hypoxic *Hif1α⁻/⁻* T cells leads to impaired AICD as well as defective cell proliferation, interestingly, it is the impaired AICD but not proliferation defect in *Hif1α⁻/⁻* hypoxic T cells that contribute to the reduced IFN-γ induction. Attenuated AICD in *Hif1α⁻/⁻* hypoxic T cells is not mediated by extrinsic factors/nutritional states but rather by cell-intrinsic alterations, that is depletion of [acetyl-CoA]. Of note, AICD has long been considered as a housekeeping process to eliminate obsolete effector T cells after a successful immune response, to save space for useful T cells and maintain immune homeostasis. AICD inhibition can lead to pathologies such as autoimmunity[50] and loss of transplantation tolerance[51]. We demonstrate here that AICD also plays an active role in shaping effector functions of hypoxic T cells, regulated by the HIF1α-glycolysis-acetyl-CoA axis. Given the important role of the Fas-FasL axis[50] in driving AICD of T cells, it would be pertinent to understand whether and how the HIF1α-glycolysis axis crosstalks with the Fas-FasL axis in hypoxic T cells.

ICBs have induced unprecedented clinical successes in various types of late-stage cancer, propelling immunotherapy as a main pillar of cancer care[54–59]. However, the overall efficacy of ICBs has reached a plateau[61], largely because of therapeutic resistance to ICBs[61,63,69]. Given that functional rejuvenation of TILs by ICBs[43,45,60] is a major mechanism underlying ICB efficacy and that cellular metabolism is the

fundamental driving force of T cell activation and effector functions[5–17], fine tuning metabolic activities to bolster effector functions of TILs has been actively pursued to boost ICB efficacy. Unfortunately, TILs and tumor cells utilize the same metabolic pathways for their growth and function, and co-live in the metabolically harsh TME characterized by hypoxia and poor nutrition, placing them in a fierce metabolic tug-of-war. How to tilt this metabolic battle to favor TILs would be key. We show that acetate supplementation can restore IFN-γ production in $Hif1\alpha^{-/-}$ TILs and overcome ICB resistance derived from HIF1α loss in T cells. Interestingly, a recent study[70] reports that acetate, by replenishing [acetyl-CoA], reverses Warburg effect in tumor cells and suppresses tumorigenesis. Furthermore, forcing tumor cells from acetate consumers to producers frees acetate in the TME, which then fuel TILs to promote anti-tumor responses[64]. Together, these studies indicate that acetate supplementation can be an ideal "two birds, one stone" strategy to concomitantly boost effector functions in TILs and suppress tumor cells.

In conclusion, we show that T cell-intrinsic HIF1α signaling, by maintaining glycolysis and [acetyl-CoA], engages AICD and drives IFN-γ induction. Loss of HIF1α in T cells impairs their tumor killing capacity and renders tumor-bearing mice resistant to ICBs, which can be overcome by acetate supplementation both in vitro and in vivo (Fig. S7E). Given that acetate supplementation (i.e., GTA) is approved by the FDA to treat infants with Canavan disease, we envision a smooth translation of our findings, which can lead to a rapid repurposing of GTA as an effective therapeutic for ICB resistance, a pressing unmet medical need.

## Methods

### Mice and cell lines

The $Hif1\alpha^{-/-}$ mice with specific HIF1α deletion in T cells were generated by crossing floxed $Hif1\alpha$ mice (Jackson Laboratory, Stock No.: 007561, B6.129-$Hif1\alpha^{tm3Rsjo}$/J) with CD4-Cre transgenic mice (Jackson Laboratory, Stock No.: 022071, B6.Cg-Tg(Cd4-cre)1Cwi/BfluJ)[7]. $Vhl^{-/-}$ mice with VHL specifically deleted in T cells were generated by crossing floxed $Vhl$ mice (Jackson Laboratory, Stock No.: 012933, B6.129S4(C)-$Vhl^{tm1Jae}$/J) with CD4-Cre transgenic mice. CD45.1 mice on B6 background (Jackson Laboratory, Stock No.: 002014, B6.SJL-$Ptprc^a$ $Pepc^b$/BoyJ) were procured from The Jackson Laboratory (Bar Harbor, ME) and bred in our animal facility. All mice were housed in specific pathogen-free conditions in the animal facility of The University of Alabama at Birmingham (UAB) under 12 h/12 h light/dark cycle, ambient room temperature (22 °C), and 40%–70% humidity. Seven to twelve-week-old mice were used in the experiments. Male mice were used for MB49 model, as it was generated in a male mouse to avoid cross-sex immune response. To facilitate random assignment of mice inoculated with B16-BL6 melanoma cells to different groups, we used female mice, as adult male mice typically show aggressive behaviors and could confound the experiments with uneven distribution of tumor sizes. Our animal protocol was approved by Institutional Animal Care and Use Committee at UAB. All tumor-bearing mice were humanely euthanized prior to tumors reaching the maximally allowed tumor size (20 mm in diameter), as indicated in our animal protocol. MB49 cells were kindly provided by Dr. A. Kamat at MD Anderson Cancer Center and cultured in DMEM supplemented with 10% FBS and 100 units/mL of penicillin and 100 μg/mL of streptomycin (all from Invitrogen). B16-BL6 cells were kindly provided by Dr I. Fidler at MD Anderson Cancer Center and cultured with MEM supplemented with 10% FBS, 2mM L-glutamine, 1 mM sodium pyruvate, 1% non-essential amino acids, 1% vitamin, 100 units/mL of penicillin and 100 μg/mL of streptomycin (all from Invitrogen). All cells were cultured in a humidified 37 °C incubator with 5% $CO_2$. All cells were regularly evaluated with the MycoAlert detection kit (Lonza, LT07-118) to ensure they were free of mycoplasma contamination.

### Mouse naïve T cell isolation, activation, and treatments

Naïve T cells were isolated from spleens and lymph nodes by negative selection using microbeads and following the manufacturer's instructions (Miltenyi, #130-104-453 for CD4+ T cells and #130-096-543 for CD8+ T cells). The purity of naïve CD4+ CD62LhiCD44loCD25− and CD8+ CD62LhiCD44lo cells were confirmed by flow cytometry. Freshly isolated naïve T cells were stimulated with plate-bound anti-CD3 (Clone 145-2C11, Bio X cell, #BE0001-1) and anti-CD28 (Clone 37.51, Bio X cell, #BE0015-1) in Click's medium (plus β-mercaptoethanol) (Irvine Scientific, #9195-500 mL) supplemented with 10% (vol/vol) FBS, antibiotics, and 100 U/mL human IL-2, in the presence or absence of IL-12 (0.2 ng/mL). Plates were pre-coated with 2 μg/mL anti-CD3 and anti-CD28 for at least 2 h. For hypoxic condition, cultures were placed in a hypoxic chamber with oxygen level set at 1% to mimic the $O_2$ tension in the TME of solid tumors. For some experiments, an oxygen level of 2.5% was used. Where designated, 50% of old media were replaced with freshly prepared Click's medium daily. As indicated, 20 mM of NaAc, 1 mM and 5 mM of β-hydroxybutyrate (BHB, Sigma, #298360), 0.5 μM of 2-DG (Sigma, #8375), or solvent controls were added to the culture medium. For glucose consumption, culture medium of activated CD4+ T cells was collected to determining glucose concentration by a Glucose Colorimetric Detection Kit (Invitrogen™, EIAGLUC). Medium alone in wells of the same plate was used as controls. The glucose consumption was calculated by subtracting the remaining glucose in the culture medium. For flow cytometry–based glucose uptake assays, activated CD4+ T cells were collected and incubated with 100 μM 2-NBDG (Invitrogen™, N13195) for 30 min before measuring fluorescence by flow cytometry.

For in vitro CD45.1/CD45.2 co-culture, naïve CD45.1+ CD4+ T cells were equally mixed with naïve WT or $Hif1\alpha^{-/-}$ CD4+ T cells (congenically marked by CD45.2) and similarly activated as above. To block IL-2, anti-mouse IL-2 (Bio X Cell, #BE0043) were added at 10 μg/mL. To assess T cell proliferation, naïve T cells were pre-labeled with 2 μM CellTrace Violet (CTV, Thermo Fisher, #C34557) by incubating for 20 min with periodical mixing. After incubation, cells were washed twice with complete culture medium to remove soluble CTV. In the co-cultures of hypoxic T cells with MB49 cells, naïve WT and $Hif1\alpha^{-/-}$ T cells were activated for 5 days under hypoxia to harvest live cells, which were then co-incubated with MB49 tumor cells at the ratio of T cells (effectors): MB49 cells (targets) of 2:1. The co-culture plate was pre-coated with 0.2 μg/mL of anti-CD3 for 2 h. Cell apoptosis of tumor cells and IFN-γ production in T cells was analyzed 48 h later.

### Human naïve CD4+ T cell isolation and activation, 2-DG treatment

Human naïve CD4+ T cells were isolated from Leukocyte Reduction System (LRS) Cones (procured from Lifesouth community blood centers). Total PBMC cells were flushed from human LRS cones with phosphate-buffered saline (PBS) containing 2% fetal bovine serum (FBS), spun at $800 \times g$ for 10 minutes, and then lysed red blood cells. Naïve CD4+ T cells were purified by negative selection using microbeads and following the manufacturer's manual (Miltenyi, #130-094-131). The purity was more than 95% determined by flow cytometry. Freshly isolated naïve CD4+ T cells were activated with plate-bound anti-human CD3 (Clone UCHT1, BioLegend, #300438) and anti-human CD28 (Clone CD28.2, BioLegend, #302934) in Click's medium (plus β-mercaptoethanol) supplemented with 10% (vol/vol) FBS and antibiotics, in the presence of 100 U/mL human IL-2. In some experiments, cultured cells were treated with 0.5 μM of 2-DG or solvent control.

### Plasmid construction, virus packaging and transduction

The coding sequences (CDS) of murine $Hif1\alpha$, $Glut1$, $Pkm2$, $Ldha$ and $Mct4$ were PCR-amplified from mouse first-strand cDNA library produced by reverse transcription (Invitrogen, #11752-050) and cloned

into the retroviral vector pMIG II. The inserts were validated by DNA sequencing. Retroviruses were produced by co-transfection of phoenix cells with transfer plasmid and packaging plasmid pCL-Eco. Virus-containing culture medium was collected at 48 h and 72 h post-transfection. To knock down human *HIF1A*, we designed and synthesized the oligoes from IDT and cloned it to retroviral vector LMP. The target sequence of human sh*HIF1A* is GGGTTGAAACTCAAGCAACTG. To knock down mouse *Hif2α*, we designed and synthesized the oligoes from IDT and cloned it to retroviral vector LMP. The target sequence is AGGAAAGCTTTGGCGTCATTC. The inserts were validated by DNA sequencing. Retroviruses expressing shRNAs were packaged by co-transfection of transfer plasmid and packaging plasmid pMD2.G to phoenix cells. Retroviral particles were then used to transduce activated human and mouse T cells using spin-infection approach (described below). Freshly isolated naïve CD4$^+$ T cells were activated overnight before retroviral spin-infection in virus-containing culture medium with the addition of Lipofectamine 3000 (Invitrogen, #L3000-150) and human IL-2, as we previously described[71]. Successfully transduced T cells expressing GFP were sorted by flow cytometry. The overexpression and knockdown of interested genes was validated by Western blot or real-time RT-PCR. To check IFN-γ production, sorted GFP$^+$ cells were re-activated on 0.2 μg/mL anti-CD3 pre-coated plate for 4 days under hypoxia.

## Cell apoptosis assay

Cell apoptosis was analyzed by flow cytometry, as we previously described[72]. Cells were harvested at the designated times, washed once with DPBS and again with 1x Annexin V binding buffer before staining with Annexin V (1:50, Thermo Fisher, #17-8007) and 7-AAD (1:200, Sigma, #129935) or propidium iodide (1:50, Thermo Fisher, #00-6990-42) in 1x Annexin V binding buffer for 30 min at room temperature. Cells were acquired by Attune NxT Flow Cytometer and data were analyzed by FlowJo. For anti-apoptotic treatment of T cells, T cells were activated by plate-bound anti-mouse CD3 and CD28 in the presence of 50 μM of caspase inhibitor z-VAD-fmk (Promega, #G7232). Apoptosis was analyzed on Day 3.

## In vivo tumor inoculation and treatment

Mice were shaved on the right flank one day before tumor inoculation. On Day 0, anesthetized mice were inoculated subcutaneously with $5 \times 10^5$ of MB49 cells to the right flank. Anti-CTLA-4 (Bio X Cell, clone 9H10) and anti-PD-1 (Bio X Cell, clone 29 F.1A12) or isotype controls was given by intraperitoneal injection (i.p.) on Day 6, 9, and 12 at a dose of 200, 100, and 100 μg per mouse, as we previously described[43]. To establish orthotopic B16-BL6 melanoma model, anesthetized mice were inoculated intradermally with $1.25 \times 10^5$ of B16-BL6 cells into the shaved right flanks on Day 0. Tumors were measured by caliper every other day starting from Day 6 and tumor volumes (mm$^3$) were calculated using the formula $(0.52 \times \text{length} \times \text{width}^2)$. The tumor-bearing mice were sacrificed at the designated time points. Upon euthanization, tumors, draining lymph nodes (DLNs) and spleens were collected, and tumor weights were recorded. For sodium acetate (NaAc) treatment in vivo, mice were treated with 500 mg/kg i.p. plus drinking water containing 200 mM NaAc starting from Day 3. Control mice were injected with equal volume of DPBS at the same time and fed with regular drinking water. For 2-DG treatment in vivo, mice were given drinking water containing 0.4% w/v of 2-DG, 2 days after tumor inoculation. Control mice were fed with regular drinking water.

## TILs isolation, tumor-draining lymph nodes (DLN) and splenocyte preparation

Tumors were collected into ice-cold RPMI 1640 containing 2% FBS and minced into fine pieces on ice, followed by digestion with 400 U/mL collagenase D (Worthington Biochemical Corporation, #LS004186) and 20 μg/mL DNase I (Sigma, #10104159001) at 37 °C for 40 min with

periodic mixing. EDTA (Sigma, #1233508) was then added to the final concentration of 10 mM to stop digestion. Cell suspensions were filtered through 70 μM cell strainers, and TILs were obtained by collecting the cells in the interphase after Ficoll (MP Biomedicals, #091692254) separation. Spleens and DLNs were collected in ice-cold HBSS containing 2% FBS to prepare single cell suspensions. Cells were filtered through 70 μM nylon mesh after lysis of red blood cells. TILs, DLN cells, and splenocytes were all re-suspended in complete Click's medium (Irvine Scientific, #9195-500 mL) for subsequent staining and flow cytometric analyzes.

## Flow cytometric analysis

For surface staining, single cell suspensions were incubated with antibody cocktails in DPBS containing 2% (wt/vol) BSA for 30 min on ice. LIVE/DEAD™ Fixable Aqua Dead Cell Stain Kit (Thermo Fisher: L34957) was performed according to the manufacturer's instructions. To stain transcriptional factors, cells were fixed by the fixation buffer in the FoxP3/Transcription Factor Staining Buffer Set (Invitrogen, #00-5523-00) after surface staining, and then stained intracellularly according to the manufacturer's instructions. To detect intracellular cytokines, cells were briefly stimulated for 4–5 h with phorbol 12-myristate 13-acetate (PMA, final concentration: 50 ng/mL; Sigma, #P8139-5MG) plus ionomycin (final concentration: 1 μM; Sigma, #I0634-1MG) in the presence of monensin (BD Biosciences, #51-2092KZ). Stimulated cells were stained with surface markers first, then fixed using the fixation buffer in the BD Cytofix/Cytoperm Plus Fixation/Permeabilization Kit (BD Biosciences, #554715), and intracellularly stained for cytokines with corresponding antibodies, according to the manufacturer's instructions. Used flow antibodies are listed in the Supplementary Table 1. All the flow cytometric data were acquired using the built-in software of the Attune NxT Flow Cytometer (Invitrogen, A24860) from Thermo Fisher. Flow cytometric data were analyzed using FlowJo (version 10.8.1), and main gating strategies were provided in Fig. S8.

## Western blot (WB) analysis

Western blot was performed, as previously described[73]. Briefly, cells were washed with cold DPBS twice before lysed with M-PER buffer (Thermo Scientific, #78501) containing proteinase inhibitors cOmplete (Roche, #11836170001) and phosphatase inhibitors (Sigma, #P2850 and P5726). Lysates were then collected and transferred to 1.5 mL Eppendorf tubes and sonicated. Protein concentration was determined by NanoDrop. Fifty μg of total proteins were loaded onto each lane of an 8-12% SDS-PAGE gel. After electrophoresis, proteins on the gel were transferred to 0.45 μm of PVDF membrane (Millipore, #IPVH00005) in a sponge sandwich. Membranes were then blocked with 5% of non-fat milk (Bio-Rad, #170-6404) and probed with primary antibodies overnight on a shaker in cold room. Membranes were then washed and incubated with HRP-conjugated secondary antibodies for 1 h at room temperature. The membranes were then incubated with Western HRP substrate (Millipore, WBLUR0500) for 2-5 min before imaging with an X-ray film. The antibodies used for WB are listed in the Supplementary Table 2. β-actin was blot as a loading control on the same gel with proteins of interest. Uncropped and unprocessed scans of all blots were provided in the Source Data file.

## Acetyl-CoA measurement

Naïve CD4$^+$ T cells were differentiated under hypoxia for 4 days before cultured in fresh Click's medium for another 24 h with or without 20 mM NaAc. Cells were lysed with M-PER buffer for 10 minutes on ice. Cell lysates were spun down at 13000 rpm for 10 min at 4 °C. Supernatant was deproteinized using 4 M perchloric acid and neutralized by 1 N potassium hydroxide. Then deproteinized lysates were used for acetyl-CoA measurement using Acetyl-Coenzyme A Assay Kit (Sigma, #MAK039) following the manufacturer's instructions. Fluorescence

intensity was measured at Ex/Em=535/587 nM. Acetyl-CoA levels were normalized to cell number.

## Real-time RT-PCR

Total RNAs were extracted from cells using RNeasy Plus Mini kit (QIAGEN, #74136) according to the manufacturer's instructions. Reverse transcription polymerase chain reaction (RT-PCR) was done, as we previously described[52]. In brief, first-strand cDNAs were synthesized by SuperScript III reverse transcriptase (Invitrogen, # 11752250) according to the manufacturer's instructions. Up to 1 µg of total RNA was used for reverse transcription. First-strand cDNA was diluted 20 times for quantitative RT-PCR which was performed on Bio-Rad CFX96 instrument. The primers are listed in the Supplementary Table 3. β-actin was used as the housekeeping gene. Specificity of primers were all validated by single peak of melting curve. The gene expression level was calculated using the $2^{-\Delta\Delta CT}$ method.

## RNA-seq analysis

Naïve CD4$^+$ T cells isolated from WT or $Hif1\alpha^{-/-}$ mice were activated for 48 h. Cells were directly lysed on the plate and total RNA was extracted immediately by RNeasy Plus Mini Kit from QIAGEN, Inc. Standard RNA-seq was performed by GENEWIZ, Inc. Briefly, total RNA was enriched with Poly A selection and sequencing was performed on Illumina platform. For RNA-seq data analysis, paired-end transcriptome sequences were mapped to the *Mus musculus* GRCm38 reference genome available on ENSEMBL using the STAR aligner (version 2.7.5a). Read counts per gene were calculated using htseq-count in the HTseq package (version 0.11.2)[74] and used for downstream differential gene expression analysis and pathway enrichment analysis. Analysis of differentially expressed genes (DEGs) between WT control and $Hif1\alpha^{-/-}$ samples was performed using DESeq2 (version 1.34.0)[75] in R (version 3.6.0). The Wald test was used to calculate the p-values and log2 fold changes. Genes with an adjusted p-value < 0.05 and absolute log2 fold change >1 were considered as DEGs. A volcano plot was used to show all upregulated and downregulated DEGs using the ggplot2 package (version 3.3.6) (ggplot2: Elegant Graphics for Data Analysis. Springer-Verlag New York. ISBN 978-3-319-24277-4, https://ggplot2.tidyverse.org). Enriched Kyoto Encyclopedia of Genes and Genomes (KEGG) pathways[76] of the DEGs were identified by enrichr package[77] (version 3.0), a comprehensive gene set enrichment analysis tool. Significant terms of the KEGG pathways were selected with a p-value < 0.05.

## CUT&RUN analysis

CUT&RUN reactions were conducted, following manufacturer's instructions (Epicypher 14-1048). In brief, naïve CD4$^+$ T cells were activated under hypoxia for 2 days and subsequently cultured in fresh Click's medium for an additional 2 days, with or without 20 mM sodium acetate. Following activation, either 50000 or 100000 activated T cells prepared by Ficoll separation were bound to activated Concanavalin A (ConA) beads and incubated overnight at 4 °C with specific antibodies against H3K9Ac or control IgG isotype in antibody buffer. Samples were washed with Digitonin buffer to permeabilize cells and prepare them for MNase (micrococcal nuclease) digestion. Subsequently, pAG-MNase enzyme (fused MNase to proteins A and G) was added to the samples and incubated at room temperature. Following MNase activation with CaCl$_2$, antibody-bound chromatin was digested, and the reaction was stopped after 2-h incubation at 4 °C with Stop buffer containing spike-in *E. coli* DNA. Antibody-bound chromatin fragments were released, and digested DNA was eluted and used for further library preparation. Libraries were prepared using NEBNext Ultra II DNA Library prep kit following the manufacturer instructions. Paired-end next-generation sequencing was conducted by Novogene Corporation, CA. Subsequent bioinformatics analysis aligned sequenced reads to the mouse mm10 genome, and peaks of protein-DNA binding were visualized using the Integrative Genomics Viewer (IGV).

## Seahorse metabolic flux assay

Oxygen consumption rates (OCR) and extracellular acidification rate (ECAR) were assessed using a Seahorse XFe96 Analyzer. WT or $Hif1\alpha^{-/-}$ naïve CD4$^+$ T cells were activated in a hypoxic chamber (1% O$_2$) for 24 h and then incubated with or without 20 mM sodium acetate for another 24 h. Subsequently, $2 \times 10^5$ T cells/well were plated on Cell-Tak coated Seahorse 96-well plate. After a quick spin, cells were allowed to settle down for 1 h. The cells were cultured in XF medium, a non-buffered RPMI-1640 medium supplemented with 5 mM glucose, 2 mM glutamine, and 1 mM sodium pyruvate, with adjusted pH value of 7.4. OCR and ECAR were measured using the Seahorse XFe96 Analyzer. OCR and ECAR values at the baseline and after injections of Oligomycin (1 µM), FCCP (1.5 µM), as well as Antimycin A (1 µM)+Rotenone (Rot/Anti) (100 nM) were recorded periodically.

## $^{13}$C-glucose metabolic flux assay and analysis

Wild-type and $Hif1\alpha^{-/-}$ naïve CD4$^+$ T cells were activated in Click's medium (containing 5 mM glucose) overnight, then supplemented with 15 mM of [U-13C]glucose (Cambridge Isotope Laboratories, Inc., Catlog #: CLM-1396-1). To quantify concentrations of metabolites in spent media, supernatants were collected at 0 (immediately after adding [U-13C]glucose), 6, and 24 h from three biological replicate cultures. These supernatants were then derivatized using MOX-TBDMS and analyzed by GC-MS as we previously described[78]. Metabolite uptake and secretion rates were then calculated[79]. For GC-MS analysis of isotopic labeling of intracellular metabolites, dried cell extracts were prepared from harvested cell pellets at 6 and 24 h after washing with cold DPBC (4 °C) once and similarly derivatized[78]. The identity of the measured metabolites was verified using pure analytical standards as described by us[80]. To verify the labeling of glucose in the medium, we followed the protocol described by Long et al.[81]. All GC-MS analyzes were performed on an Agilent 7890 A GC system equipped with a HP-5MS capillary column (30 m, 0.25 mm *i.d.*, 0.25 µm-phase thickness; Agilent J&W Scientific), connected to an Agilent 5977B Mass Spectrometer operating under ionization by electron impact (EI) at 70 eV. Helium flow was maintained at 1 mL/min. The source temperature was maintained at 230 °C, the MS quad temperature at 150 °C, the interface temperature at 280 °C, and the inlet temperature at 280 °C. Mass spectra were recorded in single ion monitoring (SIM) mode with 4 ms dwell time on each ion. Mass isotopomer distributions were obtained by integration of ion chromatograms[82].

Intracellular metabolic fluxes were determined by fitting extracellular uptake and secretion rates (glucose, glutamine, pyruvate, and lactate) and the measured mass isotopomer distributions of intracellular metabolites (pyruvate, lactate, alanine, malate, aspartate, glutamate, glutamine, citrate, glycerol-3-phosphate, phosphoenolpyruvate, and 3-phosphoglycerate) to a compartmentalized metabolic network model using the Metran software[80] (high-resolution $^{13}$C metabolic flux model-$^{13}$C-MFA). This model contains three distinct metabolic compartments: extracellular, cytosol, and mitochondrion. The metabolites pyruvate, citrate, malate, alanine, glutamate, and aspartate are metabolically active in both the cytosol and mitochondrion. During the metabolite extraction process, intracellular pools of metabolites are homogenized. As such, the measured isotopic labeling of these metabolites reflects the mixture of distinct metabolic pools. In the $^{13}$C-MFA model, we included mixing reactions to account for mixing of mitochondrial and cytosolic metabolite pools during the extraction[83]. The model also accounts for dilution of intracellular metabolite labeling due to incorporation of unlabeled CO$_2$ and exchange of intracellular and extracellular pyruvate. For $^{13}$C-metabolic flux analysis, data from three biological replicate tracer experiments were fitted to the $^{13}$C-MFA model to estimate intracellular fluxes. To ensure that the

global best solution was obtained, flux estimation was repeated at least 50 times starting with random initial values. At convergence, a chi-square test was applied to test the goodness-of-fit, and accurate 95% confidence intervals were calculated by determining the sensitivity of the sum of squared residuals to flux parameter variations[84]. The measured extracellular rates and isotopic labeling data used for $^{13}$C-MFA and the complete $^{13}$C-flux results are provided in supplementary materials.

## Statistical analysis

For animal experiments, 5–7 mice were included in each group; for in vitro studies with cells, triplicates were set up to ensure consistency and reproducibility. All experiments were repeated 2–5 times. Representative results were expressed as mean ± SEM. Data were analyzed using a two-sided Student's t-test, one-way ANOVA, or two-way ANOVA after confirming their normal distribution. All analyzes were performed using Prism 9.4.0 (GraphPad Software, Inc.) and $p < 0.05$ was considered statistically significant.

## Reporting summary

Further information on research design is available in the Nature Portfolio Reporting Summary linked to this article.

## Data availability

The RNA-seq and CUT&RUN data generated in this study have been deposited in the Gene Expression Omnibus (GEO) database under accession code GSE 253090 https://www.ncbi.nlm.nih.gov/bioproject/?term=GSE253090 Deposited data are publicly available. The remaining data in this study are available within the manuscript or Supplementary Data, with source data provided herein. Source data are provided with this paper.

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

## Acknowledgements

We acknowledge other members of Shi lab and the Department of Radiation Oncology at UAB for their constructive input. We are grateful for the Startup fund from the Department of Radiation Oncology and the O'Neal Invests pre-R01 Grant from the UAB-O'Neal Comprehensive Cancer Center granted to Shi lab. This study is also funded by National Institutes of Health grants (1R21CA230475-01A1, 1R21CA259721-01A1, and 1R01CA279849-01A1), the V Foundation Scholar Award (V2018-023), a DoD-Congressionally Directed Medical Research Programs grant (ME210108), a Cancer Research Institute CLIP Grant (CRI4342) to L.Z.S.

## Author contributions

H.S. designed and did the experiments with cells and mice, analyzed data and contributed to writing the manuscript; O.A.O., H.D., L.M., A.Y., V.Y.S., Z.L., and E.P. performed the experiments with cells and/or mice; O.A.O. analyzed CUT&RUN data; C.X. performed the bioinformatic analyzes of RNA-seq data; S.A.A. and M.I.H. assisted with the Seahorse assay and data analysis; M.R.A. did $^{13}$C-glucose metabolic flux and analyzed the data; J.A.B. contributed to manuscript construction and discussion; L.Z.S. was responsible for the original conceptualization of this study, overall data presentation, and manuscript construction; L.Z.S. acquired funding for this study, designed experiments, supervised laboratory studies and data analyzes, wrote and edited the manuscript. All authors have met the requirements for authorship and are in consensus on the content of this publication.

## Competing interests

The authors declare no competing interests.
