## [Peer Review file · Nature Communications]

HIF1 α -regulated glycolysis promotes activation-induced cell death and IFN- γ induction in hypoxic T cells

Corresponding Author: Professor Lewis Shi

Version 0:

Reviewer comments:

Reviewer #1

(Remarks to the Author)

Hypoxia is involved in various pathological settings and influences T-cell differentiation, and HIF1 α is the primary regulator of cellular adaptive responses to hypoxia. In this manuscript, "HIF1 α -glycolysis engages activation-induced cell death 1 to drive IFN- γ induction in hypoxic T cells", Shen et al. have illustrated that HIF1 α -glycolysis regulates IFN- γ induction in T cells activated under hypoxia, thereby acting as critical regulator in anti-tumor immunity. They found that T cell-specific HIF1 α deletion or glycolytic inhibition reduces IFN- γ induction and impaired tumor-killing ability, primarily due to attenuated activation-induced cell death. Meanwhile, treatment of acetate in hypoxic T-cell specific HIF1 α ^{-/-} restores IFN- γ production and re-sensitises T-cell specific HIF1 α tumor-bearing mice to ICBs. This manuscript addresses a timely topic and makes a relevant contribution to the field. However, some major revisions are needed before it can be published.

My comments are as follows:

-Various studies, for instance, <https://doi.org/10.4049/jimmunol.1402552>, and <https://doi.org/10.1002/eji.202250182>, <https://doi.org/10.1038/cddis.2017.235>, have previously reported that HIF1 α play a negative role in the IFN- γ secreting CD4⁺ T cells and an oncogenic role, especially in the TME, where hypoxia, along with elevated HIF1 α , acts as a key factor in facilitating the tumor progression, aggressiveness, and metastatic. However, this raises the question of whether there is a paradoxical nature of the hypoxic TME; how does it simultaneously facilitate tumor growth and activate anti-tumor immune responses?

-How come in Fig 2A-B, HIF1 α TM overexpressed HIF1 α but it reduced IFN γ ⁺ cells in WT compared to EV. Also, in Fig 2A, why WT HIF1 α didn't overexpress HIF1 α in WT T cells?

-Meanwhile, authors have shown that 2-DG decreases IFN- γ secretion. Based on this, it should be linked with tumor progression. However, <https://doi.org/10.1016/j.ejca.2019.09.005> showed that 2-DG reduced tumor growth. It is recommended to supplement the data showing the tumor growth and IFN- γ TILs in 2-DG treated WT and HIF1 α ^{-/-} mice.

-Likewise, tumor growth and IFN- γ -secreting TILs should be shown in NaAc-treated tumor-bearing mice.

-It is suggested to supplement the data showing the secretion of IFN- γ in WT and T-cell specific HIF1 α ^{-/-} mice at various days, i.e., day 1 to day 5. Especially when in line 227-228, the authors mentioned IFN γ is associated with the late stage of T cell activation and barely detectable on day 2-3. However, it exerted tumor-killing effects on day 2.

-in lines 173, 174, and so on, the authors have mentioned the numbers for KEGG pathways shown in figures. To enhance clarity and ease of reference, it is recommended to either replace these numerical identifiers with the actual names of the pathways or to include these numbers alongside the pathway names in the figures.

-line 189-190, mRNA expression was analysed in naïve or activated T cells?

In many places, authors have mentioned cells were activated for 2 days or 5 days. But in figure legends, they have mentioned 2.5 or 5.5 days, which is confusing.

-Would the increased number of live cells from HIF1 α ^{-/-} mice or 2-DG treated cells play some role as T memory cells in vivo?

-Fig 3A, hypoxia elevated CD25. Did authors check whether T-specific HIF1 α ^{-/-} affects the Treg differentiation and function?

-In apoptotic assay, please define if authors have used both early and late (necrotic)-stage apoptotic cells for the bar graphs.

-MS needs careful proofreading as there are several errors, like in line 570, "two birds, on stone", only naïve WT cells are mentioned in line 1133, etc.

-In Fig S1A, the bar graph with WT +IL-12 group doesn't match the values shown in FACS.

-In Fig 2 and S2, actin should be b-actin.

Reviewer #2

(Remarks to the Author)

Summary:

The authors explore the theme of Hif1a-mediated control of glycolysis in the T cells. Authors apply the well-established concept that Hif1a regulates the expression of key metabolic genes as it allows for adaptation to low oxygen environment by using Hif1a-deficient T cells. Unsurprisingly, these T cells appear to be defective in hypoxia as the cells are largely unable to adapt to 1% O₂. The authors then focus on the defective expression of IFN γ in these cells and conclude that IFN γ is a specific target of Hif1a-promoted aerobic glycolysis. Furthermore, the researchers apply previously published findings that indicate T cell activity in hypoxic environment can be augmented by providing a glycolysis independent metabolic substrate in the form of acetate. Addition of acetate to Hif1a-deficient cells rescues their ability to produce IFN γ and to promote killing of tumor cells *in vitro* and augments their ability to kill tumors *in vivo*. While studying these phenomena, authors arrive at the surprising conclusion that AICD is necessary for IFN γ production in T cells. The augmentation of Hif1a activity by acetate is then employed in an *in vitro* and *in vivo* tumor killing study. Overall, the study has many strong points, but there are several general and specific problems with the work as presented.

General points:

The Hif1a mediated control of glycolysis and T cell metabolism, the application of acetate to augment T cell activity in the hypoxic environment of the tumor and the metabolic dependence of T cell on alternate energy sources in hypoxia is well-described in literature. The authors apply the known concepts to Hif1a-deficient T cells and observe a very predictable metabolic deficiency in these T cells which are unable to adapt to hypoxia. While it is commendable to undertake the study, it provides little novel insight into the biology of T cells.

The RNA sequencing data confirms this point, providing for mostly known and characterized hypoxia target genes.

Major comments:

1. IFN γ production is thought to be dependent on the transcriptional activity of Tbet. The evidence for Tbet activation is only provided for one timepoint (day 2) while the study looks at IFN γ production at different timepoints and conditions. If Hif1a is indeed regulating the IFN γ production as the authors claim, then Tbet activity should mirror the defect seen with IFN γ and such evidence is missing in the study. Staining for other T cell lineage transcription factors need to be included as well to account for possible developmental differences in the Hif1a deficient T cells
 2. At the same time, while IFN γ is very important in T cell activation, an analysis of another cytokines such as TNF α is missing in this study. If the Hif1a regulation is IFN γ specific like the authors claim, then the expression of TNF α would presumably be unaffected. The authors need to address the TNF α production and the frequency of TNF α + IFN γ + cells needs to be shown. Analysis of mRNA and protein levels would be most informative.
 3. The authors use VHL mice as stabilized Hif1a controls. The VHL deletion stabilizes both Hif1a and Hif2a and the contribution of Hif2a in IFN γ induction has been described previously and needs to be addressed if these animals are used. A more suitable model to evaluate the presumed role of Hif1a in IFN γ production would be to use the LSL-HIF1dPA strain with the CD4 Cre driver and these experiments should be performed to validate the claims made by the authors.
 4. Authors use Click's media to culture T cells but the addition of b-mercaptoethanol is not noted, and considering the metabolic aspects under study, such media modifications need to be noted, especially if it is not included, as it will likely change the outcome of the T cell growth experiments.
 5. Authors analyze IFN γ production with a focus on day 5.5, while simultaneously providing evidence that a majority of T cells at that point have died, which seems counterintuitive. What happens at other time points and why is such late timepoint chosen?
 6. The authors need to elaborate on experimental differences between published reports that use naïve Hif1a^{-/-} T cells driven by the CD4 Cre (for example, Dang EV et al, Cell. 2011 Sep 2;146(5):772-84) and their very different outcome of the proliferation and IFN γ studies, especially as similar methods appear to be employed.
 - 1) The lymphoid organs appear to exhibit a range of oxygen tension levels, with HEV areas being most highly oxygenated. Therefore, the choice of 1% O₂ for the duration of T cell activation study might not completely reflect the physiological scenario and other O₂ levels need to be addressed. Are the differences of growth rates/cytokine secretion preserved at 2.5% or higher O₂ tension?
 7. How can the authors distinguish if the IFN γ production defect in Hif1a^{-/-} T cells is not simply a result of cell "starvation" for energy sources, and leads to a global impact on protein translation, as opposed to the presumed specific effect on IFN γ only?
 8. Are Hif1a-deficient cells that are treated with acetate able to catch up to WT cells with respect to proliferation and do these treated cells become responsive to IFN γ -mediated AICD?
 9. The Hif1a deficient T cells are generally thought to have developmental defects that promote their particular skewing towards specific lineages and that generally leads to increase to IFN γ production in these cells under normoxic conditions. In humans miR-182 represses HIF1a and leads to exacerbation in IFN γ -producing T cells in EAE patients. How can the author's results be reconciled with this existing evidence?
 10. Have the authors considered using other energy sources, for example ketones, that would rescue the IFN γ production defect in the Hif1a^{-/-} T cells?
 11. Authors do not show the dynamics of CD8 T cell proliferation and IFN γ production in the context of Hif1a deficiency under hypoxia. Is the IFN γ production and proliferation of these cells also defective under the conditions studied?
- Minor comments:
12. In figure 6, the final panel indicating biochemistry of induction of AICD actually reads ACID; needs to be corrected.

13. The Flow Cytometry panels need to show scales with numbers.
14. Previous published works addressing application of acetate to enhance T cell function in hypoxic/tumor sites should be acknowledged and cited (PMC10615731, PMC6544383).

Reviewer #3

(Remarks to the Author)

The majority of anatomical environments in the body are hypoxic, begging the question of T cell function under these conditions. In this contribution by Shen et al., the authors examine the role of HIF1 α -glycolysis in the induction of IFN γ under hypoxic conditions. Overall, the study is well designed, and the paper is well written. Specific suggestions for improvement of the paper are below.

Major questions to be resolved:

- 1) The mechanism by which acetate rescues the phenotype is unclear in your model. Are you saying something about complete glucose oxidation via the mitochondria, or are you implying acetyl-CoA for epigenetic changes? Both, neither? I was expecting some data such as ChIPseq or histone acetylation status to indicate how this axis is working.
- 2) Overall, there are a lot of data surrounding metabolism, but no direct measures of metabolism. Flux, gene expression and protein content do not always coincide.

Minor edits/suggestions/questions:

- 3) According to the standard conventions for genetic nomenclature, gene symbols in mice should be italicized and, for a knockout mouse, the gene symbol is typically written in lowercase. i.e., *Hif1* ^{-/-}. You are referring to the genotype of the mouse, please correct this throughout the paper.
- 4) Figures: Sample numbers (N = ##) missing for many experiments.
- 5) Figure 1D: What is the comparison that is being made here for the RNAseq? Please be clear in the figure legend.
- 6) Figures 1G and H: the regulation of glycolysis through changes in gene and protein expression is just one aspect of its control, and it often represents a longer-term adjustment rather than immediate regulation. Glycolysis is primarily regulated through allosteric regulation of key enzymes, post-translational modifications (such as phosphorylation), and changes in the levels of substrates and products, which can occur rapidly in response to cellular energy demands. Would suggest extracellular flux analysis to understand what is going on here with glycolysis. In addition, stable isotopic enrichment studies can be informative.
- 7) Do your result for figure 1/S1 suggest incomplete differentiation?
- 8) Line 242: "individual glycolytic checkpoints". The acknowledged checkpoints in glycolysis are hexokinase, phosphofructokinase, and pyruvate kinase. These checkpoints are subject to feedback regulation, where the end products of glycolysis (and further metabolic processes, like the citric acid cycle) can inhibit early steps in the pathway to prevent an overaccumulation of intermediates and maintain metabolic balance.
- 9) Line 257: "[acetyl-CoA] and found it was markedly reduced in hypoxic HIF1 α -/- T cells". Would like to see some immunoblots of PDH and phospho PDH.
- 10) Line 280: change want to wanted. Paper should be in past tense.
- 11) Line 299: "glycolytic activity": without metabolic studies, you cannot claim this.
- 12) Figure 3D: The study assumes that comparing cells at the same division state equates to comparing them at the same functional state. While this controls for proliferative history, it does not account for other cellular processes that might differ between WT and HIF1 α -/- T cells.

Version 1:

Reviewer comments:

Reviewer #1

(Remarks to the Author)

This revised manuscript by Shen et al. has been improved, and the authors have responded well to my comments. Overall, the revised version of the manuscript is more robust, and the data support the findings. I have no major concerns remaining.

However, I would suggest including a bar graph to quantify the western blot data to better represent and interpret the results.

Reviewer #2

(Remarks to the Author)

The authors have sufficiently addressed the main and specific points raised in the initial review and have satisfactorily answered pertinent questions and included new data that makes the manuscript more complete. The new data which addresses potential involvement of Hif2 α in IFN γ production is acceptable and sufficient and inclusion of the data points taken at 2.5% oxygen makes the conclusions drawn by the authors stronger.

General comments: Authors need to avoid making sweeping conclusions within the discussion section of the paper that Hif1 α is a metabolic regulator of IFN production in all hypoxic T cells as the data presented in this manuscript only indicates such regulation might be relevant in naïve T cells.

Also, authors should clarify the nomenclature used for mouse strains in the text with reference to the conditional knockouts. Finally, please reduce the number of acronyms, especially in the discussion section, to make the text more readable to a broader audience.

Reviewer #3

(Remarks to the Author)

I appreciate that the authors addressed my two main concerns.

The addition of the stable isotope data and extracellular flux analysis data really adds to what was originally mostly expression data.

The epigenetic data also adds to the proposed mechanism and strengthens the paper.

REVIEWER COMMENTS

We want to sincerely thank all three Reviewers for their insightful comments, which have greatly improved the quality of our study. Before reading our detailed point-by-point responses below, I would like to point out that we have reorganized our data into 7 figures to incorporate all the new data (Fig. 1B-C, Fig. 1F-H, Fig. S1B-D, Fig. S1H, Fig. 2F-G, Fig. S2F, Fig. 3A, Fig. 3H-I, Fig. S3D-H, Fig. S4E-G, Fig. S5B, and Fig. S7C). In our opinion, this allows a better flow of our story and prevents overwhelmingly busy figures. In addition, we have included two new supplemental tables showing metabolic flux from ¹³C-labelled glucose tracing assay (Suppl. Table 3) and CUT&RUN data on H3K9Ac expression (Suppl. Table 4) to go with Fig. 2F and Fig. 3F & S3G, respectively. Thank you in advance for re-evaluating the revision.

Reviewer #1 (Remarks to the Author) (Expertise: hypoxia signaling)

Hypoxia is involved in various pathological settings and influences T-cell differentiation, and HIF1 α is the primary regulator of cellular adaptive responses to hypoxia. In this manuscript, "HIF1 α -glycolysis engages activation-induced cell death 1 to drive IFN-g induction in hypoxic T cells", Shen et al. have illustrated that HIF1 α -glycolysis regulates IFN-g induction in T cells activated under hypoxia, thereby acting as critical regulator in anti-tumor immunity. They found that T cell-specific HIF1 α deletion or glycolytic inhibition reduces IFN-g induction and impaired tumor-killing ability, primarily due to attenuated activation-induced cell death. Meanwhile, treatment of acetate in hypoxic T-cell specific HIF1 α ^{-/-} restores IFN-g production and re-sensitises T-cell specific HIF1 α tumor-bearing mice to ICBs. This manuscript addresses a timely topic and makes a relevant contribution to the field. However, some major revisions are needed before it can be published.

We thank this Reviewer for his/her concise and positive evaluation of our study.

My comments are as follows:

-Various studies, for instance, <https://doi.org/10.4049/jimmunol.1402552>, and <https://doi.org/10.1002/eji.202250182>, <https://doi.org/10.1038/cddis.2017.235>, have previously reported that HIF1 α play a negative role in the IFN-g secreting CD4⁺ T cells and an oncogenic role, especially in the TME, where hypoxia, along with elevated HIF1 α , acts as a key factor in facilitating the tumor progression, aggressiveness, and metastatic. However, this raises the question of whether there is a paradoxical nature of the hypoxic TME; how does it simultaneously facilitate tumor growth and activate anti-tumor immune responses?

*Thanks for pointing this out. Among the three referenced studies, the **first** by Shehade, et al. used CD4⁺CD25⁻ T cells that contain ~40-50% activated effector/central memory T cells (CD44⁺) already equipped to produce IFN- γ , which would not allow an unequivocal assessment of IFN- γ induction in naïve T cells. The **second** by Wei, et al., assessed total CD8⁺ T cells stimulated with plate-bound CD3 and CD28 under normoxia but not hypoxia, which did not examine IFN- γ induction in naïve CD4⁺ T cells. The **third** by Ban, et. al. tested a HIF1 α inhibitor (IDF-11774) in nude mice that do not have T cells, which primarily assessed the effects of IDF-11774 on*

tumor cells but not T cells. To the best of our knowledge, our study is the first to explicitly examine the role of HIF1 α -glycolysis in IFN- γ induction in hypoxic T cells. While our data strongly support an essential role of HIF1 α -glycolysis axis in inducing IFN- γ production in hypoxic T cells, it is necessary to mention that this functional mechanism in T cells can be counteracted or suppressed by various immunosuppressive mechanisms in the tumor microenvironment, including myeloid-derived suppressor cells, tumor-associated M2 macrophages, abnormal tumor vasculature, and hypoxia-induced aggressive features in tumor cells (e.g., genomic instability, abnormal angiogenesis, metabolic reprogramming, cancer cell stemness, etc.). We reason that effective strategies to sustain the function of HIF1 α -glycolysis pathway in TILs would bolster anti-tumor immunity, to which acetate supplementation can be a legitimate approach.

-How come in Fig 2A-B, HIF1 α TM overexpressed HIF1 α but it reduced IFN γ ⁺ cells in WT compared to EV. Also, in Fig 2A, why WT HIF1 α didn't overexpress HIF1 α in WT T cells? *There was a decreasing trend of IFN- γ production in WT cells transduced with HIF1 α TM, but this did not reach statistical significance. We reason that the weak HIF1 α overexpression in WT T cells by WT HIF1 α (new Fig. 3A) was due to the stabilization of the endogenous WT HIF1 α . Nevertheless, there was still an increase of HIF1 α expression in transduced WT T cells, but this overexpression was substantially more evident in Hif1 α ^{-/-} T cells.*

-Meanwhile, authors have shown that 2-DG decreases IFN-g secretion. Based on this, it should be linked with tumor progression. However, <https://doi.org/10.1016/j.ejca.2019.09.005> showed that 2-DG reduced tumor growth. It is recommended to supplement the data showing the tumor growth and IFN-g TILs in 2-DG treated WT and HIF1 α ^{-/-} mice.

In the referenced study, 2-DG was administered systemically by drinking water, and the authors attributed suppression of LLC lung tumors by 2-DG to its anti-angiogenic effects in endothelial cells; there were no data on 2-DG effects on IFN- γ production in T cells. Of note, there was a greater reduction of tumor incidence and tumor growth at lower 2-DG dose (0.2% w/v) (50% reduction with tumor weights: $\sim 1.35 \pm 0.33$ g) than higher 2-DG (0.4% w/v) (17% reduction with tumor weights: $\sim 1.76 \pm 0.23$ g); it would be interesting to examine whether this was due to the suppressed TILs' effector functions by the higher dose of 2-DG. To address this Reviewer's comment, WT and Hif1 α ^{-/-} tumor-bearing mice were treated with ICB, in the absence and presence of 2-DG. Our new data (Fig. 7D and Fig. S7C). showed that 2-DG dampened therapeutic effects of ICBs, in a process dependent on T cell HIF1 α pathway, further accentuating a pivotal role of T cell HIF1 α -glycolysis in ICB efficacy.

-Likewise, tumor growth and IFN-g-secreting TILs should be shown in NaAc-treated tumor-bearing mice.

These data were included in our original submission (now, new Fig. 7E, 7F, 7G, and S7D).

-It is suggested to supplement the data showing the secretion of IFN-g in WT and T-cell specific HIF1 α ^{-/-} mice at various days, i.e., day 1 to day 5. Especially when in line 227-228, the authors mentioned IFN γ is associated with the late stage of T cell activation and barely detectable on day 2-3. However, it exerted tumor-killing effects on day 2.

We have provided new data on IFN- γ production by Hif1 α ^{-/-} CD4⁺ (Fig. 1G) and CD8⁺ (Fig. S1F) T cells on Day 1, 2, 3, 4, and 5, which showed minimal production on Day 2-3 but drastically increased on Day 4-5. To clarify and as we stated in the text, tumor killing assays were conducted with fully activated WT and Hif1 α ^{-/-} CD4⁺ T cells from Day 5, which were then cocultured with MB49 tumor cells for an additional 48 h, followed by analyses of cell death.

-in lines 173, 174, and so on, the authors have mentioned the numbers for KEGG pathways shown in figures. To enhance clarity and ease of reference, it is recommended to either replace these numerical identifiers with the actual names of the pathways or to include these numbers alongside the pathway names in the figures.

We have changed them accordingly.

-line 189-190, mRNA expression was analysed in naïve or activated T cells?

This was described in the legend (now, Fig. 2D); activated T cells for 48h were analyzed.

In many places, authors have mentioned cells were activated for 2 days or 5 days. But in figure legends, they have mentioned 2.5 or 5.5 days, which is confusing.

We have unified the time points in this revision.

-Would the increased number of live cells from HIF1 α ^{-/-} mice or 2-DG treated cells play some role as T memory cells in vivo?

The role of HIF1 α -hypoxia and 2-DG in memory T cells in vivo have been previously studied, with splitting results reported. In the studies by Liikanen, et al.¹ and by Hassan, et al.², HIF1 α and hypoxia promoted the formation of tissue-resident memory cells and antitumor response; in the study by Sukumar, et al.³, activation of CD8⁺ T cells in the presence of 2-DG enhanced the generation of memory cells and antitumor functions. We would like to emphasize that the focus of our study is on whether and how HIF1 α -glycolysis and hypoxia impact IFN- γ induction. To address this Reviewer's comment, we analyzed CD44 and CD62L expression in WT and Hif1 α ^{-/-} splenic T cells and did not observe overt changes of T_{EM} (CD44⁺CD62L⁻) and T_{CM} (CD44⁺CD62L⁺) (see data below) (Note: upon generation, memory T cells migrate to peripheral lymphoid organs, such as spleen and DLNs⁴). Based on the previous studies and our data, we reason the effects of HIF1 α deficiency and 2-DG on memory T cells are context-dependent. For this reason and our focus on IFN- γ induction, we opted not to include these new data in the revision. However, if this reviewer prefers, we would be happy to include the data as a supplemental figure.

WT and *Hif1α*^{-/-} mice were inoculated with 0.5M MB49 tumor cells, followed by ICB treatments, in the absence and presence of 2-DG (0.4% w/v drinking water, *ad libitum* for 2 weeks). Splenocytes harvested on Day 17 post-tumor inoculation was stained for CD62L and CD44 to determine the frequencies of central memory T cells (T_{CM}: CD62L⁺CD44⁺) and effector memory T cells (T_{EM}: CD62L⁻CD44⁺) in CD4⁺ and CD8⁺ T cells. Pooled results shown in the dot plots depicted means ± SEM for all the mice in each group, with each dot denoting an individual mouse.

-Fig 3A, hypoxia elevated CD25. Did authors check whether T-specific HIF1a^{-/-} affects the Treg differentiation and function?

We previously showed that HIF1α inhibited T_{reg} (our 2011 JEM). As we described in the text in our submission and this revision, “Since CD25 is a marker for T_{reg}, we stained cells for FoxP3 and observed comparable expression between WT and Hif1α^{-/-} T cells (Fig. S4C), ruling out the possibility that reduced CD25 expression in Hif1α^{-/-} T cells was due to the differential abundance of T_{reg}.”

-In apoptotic assay, please define if authors have used both early and late (necrotic)-stage apoptotic cells for the bar graphs.

Yes. We included both early and late-stage apoptotic cells.

-MS needs careful proofreading as there are several errors, like in line 570, “two birds, on stone”, only naïve WT cells are mentioned in line 1133, etc.

We apologize for these oversights and have corrected them in this revision.

-In Fig S1A, the bar graph with WT +IL-12 group doesn't match the values shown in FACS.

Thanks for pointing this out. We have updated S1A and included FACS panels and representative results from the same experiments.

-In Fig 2 and S2, actin should be b-actin.

We have changed “Actin” to “β-Actin.”

Reviewer #2 (Remarks to the Author) (Expertise: T cell metabolism/hypoxia):

Summary:

The authors explore the theme of Hif1a-mediated control of glycolysis in the T cells. Authors apply the well-established concept that Hif1a regulates the expression of key metabolic genes as it allows for adaptation to low oxygen environment by using Hif1a-deficient T cells. Unsurprisingly, these T cells appear to be defective in hypoxia as the cells are largely unable to adapt to 1% O₂. The authors then focus on the defective expression of IFN γ in these cells and conclude that IFN γ is a specific target of Hif1a-promoted aerobic glycolysis. Furthermore, the researchers apply previously published findings that indicate T cell activity in hypoxic environment can be augmented by providing a glycolysis independent metabolic substrate in the form of acetate. Addition of acetate to Hif1a-deficient cells rescues their ability to produce IFN γ and to promote killing of tumor cells in vitro and augments their ability to kill tumors in vivo. While studying these phenomena, authors arrive at the surprising conclusion that AICD is necessary for IFN γ production in T cells. The augmentation of Hif1a activity by acetate is then employed in an in vitro and in vivo tumor killing study. Overall, the study has many strong points, but there are several general and specific problems with the work as presented.

We thank this Reviewer for recognizing the strong points in our study. It was a surprising finding to us that Hif1 α ^{-/-} T cell actually survived better under hypoxia, but they were not able to be fully activated, coupled with reduced AICD and impaired IFN- γ induction. We would like to respectfully point out that there have been no prior studies assessing the role of HIF1 α -anaerobic glycolysis in IFN- γ induction in hypoxic T cells, despite the ubiquitous existence of hypoxia and the importance of IFN- γ in immunity. Our data also unveil HIF1 α as a key regulator of metabolic reprogramming in hypoxic T cells, complementing our previous finding that it is Myc but not HIF1 α that control the metabolic reprogramming in normoxic T cells.

General points:

The Hif1a mediated control of glycolysis and T cell metabolism, the application of acetate to augment T cell activity in the hypoxic environment of the tumor and the metabolic dependence of T cell on alternate energy sources in hypoxia is well-described in literature. The authors apply the known concepts to Hif1a-deficient T cells and observe a very predictable metabolic deficiency in these T cells which are unable to adapt to hypoxia. While it is commendable to undertake the study, it provides little novel insight into the biology of T cells.

We agree with this Reviewer that there have been substantial efforts assessing the role HIF1 α -glycolysis in T cell activation and function. In 2011, we published one of the first papers showing that HIF1 α -glycolysis orchestrates a key metabolic checkpoint in T_H17 and T_{reg} differentiation (2011 JEM). It is noteworthy to point out that the vast majority of previous studies focused on normoxic cell culture condition readily available in the laboratory setting (21% O₂). Moreover, both our 2011 JEM paper and the 2011 Cell paper by Dang, et al. did not show defects of IFN- γ induction and cell proliferation in Hif1 α ^{-/-} T cells, under normoxia. There has been no prior study to unequivocally assess the role of HIF1 α -glycolysis in IFN- γ induction T cells, under hypoxia. As summarized in our responses to Reviewer 1's comments (see above), previous studies used either CD4⁺CD25⁻ or total CD4⁺/CD8⁺ T cells, which contain already-activated T cells and prevent an unequivocal assessment of IFN- γ induction in naïve T cells. Our study fills this vacancy by showing that HIF1a-glycolysis is essential for IFN- γ induction in naïve T cells,

under hypoxia but not normoxia. We further show that it is AICD but not cell proliferation defect in hypoxic HIF1 α ^{-/-} T cells that underlies the impaired IFN- γ induction.

The RNA sequencing data confirms this point, providing for mostly known and characterized hypoxia target genes.

While it is understandable that glycolytic genes were among the top DEGs, there are several notable findings from our RNA-seq data, with direct confirmation with real-time RT-PCR and Western blot. As described in the text and concisely summarized here, the top enriched pathways in normoxic versus hypoxic Hif1 α ^{-/-} T cells were distinct, with just one shared pathway among the top 10 hits (i.e., the HIF1 α pathway). Most downregulated pathways in hypoxic Hif1 α ^{-/-} T cells were either directly involved or intimately interacted with cellular metabolism, suggesting HIF1 α is crucial for metabolic reprogramming in hypoxic T cells, contrasting its indispensable role in normoxic T cell activation, as shown here and in our previous 2011 Immunity paper⁵. Intriguingly, previous studies reveal an indispensable role of LDHa⁶ but not HIF1 α ^{7, 8} in IFN- γ induction in normoxic T cells; our study here on the other hand show that HIF1 α but not LDHa is required for IFN- γ induction in hypoxic T cells.

Major comments:

1. IFN γ production is thought to be dependent on the transcriptional activity of Tbet. The evidence for Tbet activation is only provided for one timepoint (day 2) while the study looks at IFN γ production at different timepoints and conditions. If Hif1a is indeed regulating the IFN γ production as the authors claim, then Tbet activity should mirror the defect seen with IFN γ and such evidence is missing in the study. Staining for other T cell lineage transcription factors need to be included as well to account for possible developmental differences in the Hif1a deficient T cells

We chose Day 2 to assess T-bet expression prior to the appreciable production of IFN- γ on day 3-5, to establish the temporal relationship of transcriptional regulation of IFN- γ induction by T-bet. Following this Reviewer's comment on "Tbet activity should mirror the defect seen with IFN γ ", we compared T-bet expression in normoxic and hypoxic Hif1 α ^{-/-} T cells, reasoning that there should be no downregulation of T-bet in normoxic Hif1 α ^{-/-} T cells, as there was no change of IFN- γ induction. Our new data confirmed this (Fig. S1B). Addressing this Reviewer's comment on "possible development difference", we assessed the expression of master TFs for other T cell lineages (e.g., Gata-3 for T_{H2}, ROR γ t for T_{H17}, and FoxP3 for T_{reg}), which did not show overt alterations in hypoxic Hif1 α ^{-/-} T cells, ruling out the possibility that altered IFN- γ production is due to the diversion to other T cell lineages (Fig. S1C).

2. At the same time, while IFN γ is very important in T cell activation, an analysis of another cytokines such as TNF α is missing in this study. If the Hif1a regulation is IFN γ specific like the authors claim, then the expression of TNF α would presumably be unaffected. The authors need to address the TNF α production and the frequency of TNF α ⁺ IFN γ ⁺ cells needs to be shown. Analysis of mRNA and protein levels would be most informative.

This is a great point. In our original submission, we included IL-2 (a cytokine associated with early stage of T cell activation⁹), which was not reduced but instead increased in hypoxic Hif1 α ^{-/-} T cells (Fig. S1G). Following this Reviewer's comment, we evaluated TNF (another early cytokine in T cell activation¹⁰), which was also significantly increased in hypoxic Hif1 α ^{-/-} T cells (Fig. S1H). But the frequency of IFN- γ ⁺TNF⁺ still appeared to be reduced (Fig. S1H), due to the severely reduced IFN- γ production. These new data support a selective requirement of HIF1 α in promoting induction of IFN- γ , GzmB, and Prf (late cytokines) but suppressing IL-2 and TNF production (early cytokines). It would be interesting in the future to delineate how HIF1 α reciprocally regulates their induction.

3. The authors use VHL mice as stabilized Hif1a controls. The VHL deletion stabilizes both Hif1a and Hif2a and the contribution of Hif2a in IFN γ induction has been described previously and needs to be addressed if these animals are used. A more suitable model to evaluate the presumed role of Hif1a in IFN γ production would be to use the LSL-HIF1dPA strain with the CD4 Cre driver and these experiments should be performed to validate the claims made by the authors.

To the best of our knowledge, the only study that reported a role of HIF2 α in IFN- γ production in T cells was the 2022 Elife paper by Ajouaou, et al.¹¹, wherein the deficiency of PHD2 (a negative regulator of HIFs) in T_{reg} led to increased IFN- γ expression, in a PHD2-HIF2 α -STAT1 dependent manner. Regarding the role of HIF1 α versus HIF2 α in IFN- γ induction in T cells, Doedens, et al.¹² (2013 Nature Immunology) showed that transferred Vhl^{-/-} P14 total CD8⁺ T cells produced more IFN- γ on Day 21 post-LCMV CL-13 infection, although the specific role of HIF1 α vs HIF2 α was not distinguished; but they did show that granzyme B induction by hypoxia was completely dependent on HIF1 α , as we reported here. Furthermore, Palazon, et al.¹³ (2017, Cancer Cell) found that CD8⁺ T cells lacking HIF1 α but not HIF2 α had reduced glycolysis and decreased IFN- γ production. Neither study evaluated the role of HIF1 α -glycolysis in IFN- γ induction in naïve T cells (both CD4⁺ and CD8⁺), and ours is the first to show an indispensable role of HIF1 α in this process. In line with a dispensable role of HIF2 α in IFN- γ production in T cells, as reported by Palazon, et al.¹³, we have provided new data showing that although hypoxia stabilized both HIF1 α and HIF2 α in T cells (Fig. 1B), only ablation of HIF1 α (Fig. 1A) but not knockdown of HIF2 α (Fig. 1C and Fig. S1D) abolished hypoxia-driven induction of IFN- γ , clearly indicating a dominant role of HIF1 α in this process.

The LSL-HIF1dPA mouse model is currently not available at Jackson Lab (<https://www.jax.org/strain/009673>) (under cryopreservation). Therefore, it will take a long time to re-derive, breed, and generate enough mice for in vivo experiments. Moreover, it is a transgenic mouse line carrying an inserted human HIF1dPA under the control of a foreign promoter (ROSA26), which lacks their native 5' and 3' UTRs and is not 100% homologous to mouse HIF1 α . Since its generation in 2006¹⁴, no studies have used this "humanized" mouse model to study T cell functions, likely because of these mismatches and potential immunological concerns. Given these facts and our aforementioned data, we hope that we can convince this Reviewer that it is HIF1 α but not HIF2 α that drives IFN- γ induction in hypoxic T cells.

4. Authors use Click's media to culture T cells but the addition of b-mercaptoethanol is not

noted, and considering the metabolic aspects under study, such media modifications need to be noted, especially if it is not included, as it will likely change the outcome of the T cell growth experiments.

We did add β -mercaptoethanol to Click's medium as we previously reported (2011 JEM⁸). We indicated this supplementation in the revision.

5. Authors analyze IFN γ production with a focus on day 5.5, while simultaneously providing evidence that a majority of T cells at that point have died, which seems counterintuitive. What happens at other time points and why is such late timepoint chosen?

Day 5/5.5 is a commonly used time point to induce fully differentiated T cells⁸. We have provided data showing IFN- γ induction from Day 1-5 (Fig. 1G and Fig. S1F), which show that IFN- γ production is more abundant on Day 4-5.

6. The authors need to elaborate on experimental differences between published reports that use naïve Hif1 α ^{-/-} T cells driven by the CD4 Cre (for example, Dang EV et al, Cell. 2011 Sep 2;146(5):772-84) and their very different outcome of the proliferation and IFN γ studies, especially as similar methods appear to be employed.

The 2011 Cell paper by Dang, et al. as well as our 2011 JEM paper assessed T_H1 differentiation/IFN- γ induction in T cells activated under normoxia but not hypoxia. Both studies showed no defect of IFN- γ induction and cell proliferation in Hif1 α ^{-/-} T cells, under normoxia. However, as we reported here, IFN- γ induction and cell proliferation are significantly impaired in hypoxic Hif1 α ^{-/-} T cells.

1) The lymphoid organs appear to exhibit a range of oxygen tension levels, with HEV areas being most highly oxygenated. Therefore, the choice of 1% O₂ for the duration of T cell activation study might not completely reflect the physiological scenario and other O₂ levels need to be addressed. Are the differences of growth rates/cytokine secretion preserved at 2.5% or higher O₂ tension?

As we stated in our original submission, we chose 1% O₂ to mimick O₂ tensions in the tumor microenvironment and inflammation sites. Following the suggestion from this Reviewer, we examined IFN- γ induction (Fig. 1F), cell proliferation (Fig. S4E), and cell death (Fig. S5B) in T cells activated under 2.5% O₂, which corroborated our results from 1% O₂, indicating a rather universal role of HIF1 α under different hypoxic conditions.

7. How can the authors distinguish if the IFN γ production defect in Hif1 α ^{-/-} T cells is not simply a result of cell “starvation” for energy sources, and leads to a global impact on protein translation, as opposed to the presumed specific effect on IFN γ only?

We found that the production of both IL-2 and TNF (protein level) was actually increased in Hif1 α ^{-/-} T cells (Fig. S1G and Fig. S1H), strongly argue that reduced IFN- γ production is not “a global impact on protein translation”. Furthermore, to rule out the “starvation” concern from this Reviewer, we replaced 50% of “old” medium with fresh “nutritious” medium daily from

Day 1-4, followed by IFN- γ examination on Day 5, which did not remedy the differences of IFN- γ production in WT versus Hif1 α ^{-/-} T cells (Fig. 1H).

8. Are Hif1a-deficient cells that are treated with acetate able to catch up to WT cells with respect to proliferation and do these treated cells become responsive to IFN γ -mediated AICD?

We analyzed cell proliferation after adding acetate and found that HIF1 α ^{-/-} T cells still showed proliferative defect (Fig. S4F). To clarify, the focus of our study is to establish impaired AICD in hypoxic Hif1 α ^{-/-} T cells is the driver but not the consequence of IFN- γ reduction; in further support of this, treating WT and Hif1 α ^{-/-} T cells with high doses of IFN- γ did not alter the phenotype (Fig. S5E).

9. The Hif1a deficient T cells are generally thought to have developmental defects that promote their particular skewing towards specific lineages and that generally leads to increase to IFN γ production in these cells under normoxic conditions. In humans miR-182 represses HIF1a and leads to exacerbation in IFN γ -producing T cells in EAE patients. How can the author's results be reconciled with this existing evidence?

Original studies by Dang et al. (2011, Cell)⁷ and us⁸ (2011 JEM) did not find overt developmental defects in thymus, spleen, and lymph nodes in mice with specific deletion of HIF1 α in T cells. Moreover, both studies did not find that IFN- γ production is impaired in Hif1 α ^{-/-} T cells, when activated under normoxia. With respect to CD8⁺ T cells, HIF1 α positively regulated IFN- γ production^{12, 13}, esp. under hypoxia (note: both studies used total CD8⁺ T cells but not naïve CD8⁺ T cells, distinct from our study). The said study on miR-182 analyzed patient samples with multiple sclerosis and T cells from experimental autoimmune encephalomyelitis mice, different from our experimental systems.

10. Have the authors considered using other energy sources, for example ketones, that would rescue the IFN γ production defect in the Hif1a^{-/-} T cells?

This is an excellent suggestion, as ketone bodies like β -hydroxybutyrate (BHB) can act as a substrate for acetyl-CoA production. A previous study reported that BHB production was impaired in CoVID-19 patients but not influenza patients with acute respiratory stress (ARS)¹⁵, and adding BHB (5 mM) to TH1 and TC1 cell cultures can increase their IFN- γ production, under normoxia. To test whether BHB supplementation, like acetate supplementation, can rescue the impaired IFN- γ induction in hypoxic Hif1 α ^{-/-} T cells, we treated hypoxic T cell with 1 and 5 mM BHB. Unlike NaAc, supplementation of BHB did not rescue the reduced IFN- γ production in Hif1 α ^{-/-} T cells (Fig. S3H), indicating a dispensable role of ketone bodies in our system.

11. Authors do not show the dynamics of CD8 T cell proliferation and IFN γ production in the context of Hif1a deficiency under hypoxia. Is the IFN γ production and proliferation of these cells also defective under the conditions studied?

We have included new data on the dynamics of CD8⁺ T cell proliferation (Fig. S4G) and IFN- γ production (Fig. S1F), which exhibit similar defects to CD4⁺ T cells, in the absence of HIF1 α .

Minor comments:

12. In figure 6, the final panel indicating biochemistry of induction of AICD actually reads ACID; needs to be corrected.

We have corrected this.

13. The Flow Cytometry panels need to show scales with numbers.

We have added scales with numbers to all the flow panels.

14. Previous published works addressing application of acetate to enhance T cell function in hypoxic/tumor sites should be acknowledged and cited (PMC10615731, PMC6544383).

We have cited both papers.

Reviewer #3 (Remarks to the Author) (Expertise: T cell metabolism/differentiation):

The majority of anatomical environments in the body are hypoxic, begging the question of T cell function under these conditions. In this contribution by Shen et al., the authors examine the role of HIF1 α -glycolysis in the induction of IFN γ under hypoxic conditions. Overall, the study is well designed, and the paper is well written. Specific suggestions for improvement of the paper are below.

We thank this Reviewer for the high evaluation of our study.

Major questions to be resolved:

1) The mechanism by which acetate rescues the phenotype is unclear in your model. Are you saying something about complete glucose oxidation via the mitochondria, or are you implying acetyl-CoA for epigenetic changes? Both, neither? I was expecting some data such as ChIPseq or histone acetylation status to indicate how this axis is working.

This is a great point. To address this, using a Seahorse Analyzer, we measured ECAR (a readout of glycolytic activity) and OCR (a readout of oxidative phosphorylation (OxPhos) through TCA cycle) of hypoxic WT and Hif1 α ^{-/-} T cells, with or without NaAc supplementation (Fig. S3D). Acetate supplementation did not substantially alter reduced glycolysis in Hif1 α ^{-/-} T cells, in line with reduced acetyl-CoA in hypoxic Hif1 α ^{-/-} T cells being a downstream consequence of impaired glycolysis. Interestingly, supplementation of NaAc inhibited both glycolysis (drastically reduced ECAR) and OxPhos (modestly reduced OCR) in both WT and Hif1 α ^{-/-} T cells (Fig. S3D), likely a negative feedback mechanism due to the restoration of [acetyl-CoA], which was further supported by reduced glucose uptake (measured by 2-NBDG uptake, Fig. S3E) and glucose consumption (Fig. S3F) by NaAc-treated T cells. To gain an idea on how NaAc epigenetically modulates chromatin structures of Ifng promoter and enhancer regions, we performed CUT&RUN analysis of H3K9Ac, a histone mark previously reported to be associated with active transcribed genes in T_H1 cells⁶. As shown in Fig. 3H, we observed increased H3K9Ac in the Ifng locus in Hif1 α ^{-/-} T cells after NaAc treatment, coupled with significantly increased

mRNA expression of Ifng (Fig. 3I). Our CUT&RUN analysis also confirmed the less active loci of Glut1 and Ldha in hypoxic Hif1 α ^{-/-} T cells (Fig. S3G), as expected.

2) Overall, there are a lot of data surrounding metabolism, but no direct measures of metabolism. Flux, gene expression and protein content do not always coincide.

We agree with this Reviewer on the potential discordance of metabolic activity to the expression of glycolytic genes/proteins. Addressing this Reviewer's comments, we assessed metabolic flux in hypoxic WT and Hif1 α ^{-/-} T cells using ¹³C-labeled glucose, which clearly showed that Hif1 α ^{-/-} T cells had significantly reduced glycolysis but increased OxPhos (Fig. 2F), coupled with reduced uptake of glucose, decreased secretion of lactate, but compensatory increase of uptake of extracellular glutamine and pyruvate (Supplemental Table 3). We also measured ECAR (a readout for glycolysis) and OCR (a readout for OxPhos) (Fig. 2G), which corroborated the metabolic flux with ¹³C-labeled glucose, in agreement with drastic reduction of glycolytic molecules in hypoxic Hif1 α ^{-/-} T cells than WT T cells (Fig. 2A-E).

Minor edits/suggestions/questions:

3) According to the standard conventions for genetic nomenclature, gene symbols in mice should be italicized and, for a knockout mouse, the gene symbol is typically written in lowercase. i.e., Hif1 α ^{-/-}. You are referring to the genotype of the mouse, please correct this throughout the paper.

We have italicized all the mouse gene names in the figures and the text.

4) Figures: Sample numbers (N = ##) missing for many experiments.

We have added sample numbers for each experiment.

5) Figure 1D: What is the comparison that is being made here for the RNAseq? Please be clear in the figure legend.

We compared the expression levels of genes in Hif1 α ^{-/-} T cells activated under hypoxia to those of their WT counterparts. We have made this point clear in the figure legends.

6) Figures 1G and H: the regulation of glycolysis through changes in gene and protein expression is just one aspect of its control, and it often represents a longer-term adjustment rather than immediate regulation. Glycolysis is primarily regulated through allosteric regulation of key enzymes, post-translational modifications (such as phosphorylation), and changes in the levels of substrates and products, which can occur rapidly in response to cellular energy demands. Would suggest extracellular flux analysis to understand what is going on here with glycolysis. In addition, stable isotopic enrichment studies can be informative.

We agree with this Reviewer and have provided new data of ECAR/OCR measurement by Seahorse XF96 Extracellular Flux Analyzer (Fig. 2G, Fig. S3D) as well as metabolic flux using ¹³C-labeled glucose (Fig. 2F), which revealed a glycolytic defect in hypoxic Hif1 α ^{-/-} T cells. This

was further confirmed by direct analyses of glucose uptake by 2-NBDG (Fig. S2F, Fig. S3E) and glucose consumption rate (Fig. S3F) in *Hif1 α ^{-/-}* T cells.

7) Do your result for figure 1/S1 suggest incomplete differentiation?

In our 2011 JEM paper, we used a T_H1 condition with irradiated antigen-presentation cells (APCs) and high doses of IL-12 that is known to induce greater IFN- γ production. However, various factors released by APCs and IL-12 can confound the effect of HIF1 α -glycolysis in IFN- γ induction in naïve T cells. As we described in the text and to limit those interfering effects from irradiated APC and IL-12, we used the plate-bound anti-CD3 and anti-CD28 condition without APCs and with limited or no IL-12 to explicitly assess the role of HIF1 α -glycolysis in IFN- γ induction. Importantly, our in vivo data corroborated our in vitro findings.

8) Line 242: “individual glycolytic checkpoints”. The acknowledged checkpoints in glycolysis are hexokinase, phosphofructokinase, and pyruvate kinase. These checkpoints are subject to feedback regulation, where the end products of glycolysis (and further metabolic processes, like the citric acid cycle) can inhibit early steps in the pathway to prevent an overaccumulation of intermediates and maintain metabolic balance.

We agree with this Reviewer on the acknowledged glycolytic checkpoints. We chose the ones with the greatest fold changes and changed our description to “metabolic molecules.”

9) Line 257: “[acetyl-CoA] and found it was markedly reduced in hypoxic HIF1 α ^{-/-} T cells”. Would like to see some immunoblots of PDH and phospho PDH.

*This is a great point. We detected PDH and p-PDH by Western blot. While we did not see overt change of total PDH, there was a marked increase of p-PDH in *Hif1 α ^{-/-}* T cells (Fig. 2E), suggesting reduced activity, consistent with decreased glycolysis in these cells.*

10) Line 280: change want to wanted. Paper should be in past tense.

We have made the change.

11) Line 299: “glycolytic activity”: without metabolic studies, you cannot claim this.

*We have provided new data on glycolytic activities (Fig. 2F-G, Fig. S2F, Fig. S3D-F), which revealed a glycolytic defect in hypoxic *Hif1 α ^{-/-}* T cells. With these new supportive data, we would like to keep this description in our manuscript. We sincerely thank this reviewer for the great comments, which have greatly improved our study.*

12) Figure 3D: The study assumes that comparing cells at the same division state equates to comparing them at the same functional state. While this controls for proliferative history, it does not account for other cellular processes that might differ between WT and HIF1 α ^{-/-} T cells.

*This is an excellent point. We analyzed WT and *Hif1 α ^{-/-}* T cells with the same division to rule out the role of cell division in IFN- γ induction in hypoxic *Hif1 α ^{-/-}* T cells. We agree with the Reviewer that this cannot account for other cellular processes underlying the proliferation defect*

but would like to point out that *Hif1* $\alpha^{-/-}$ T cells had decreased IFN- γ production compared to WT T cells regardless of the cell division. In fact, *Hif1* $\alpha^{-/-}$ T cells, even with more divisions, still produced substantially reduced amount of IFN- γ than WT cells with fewer divisions (Fig. 4D, the bar graph on the right), strongly supporting that impaired IFN- γ production was not due to defective proliferation of *Hif1* $\alpha^{-/-}$ T cells.

References

1. Liikanen I, et al. Hypoxia-inducible factor activity promotes antitumor effector function and tissue residency by CD8⁺ T cells. *J Clin Invest* **131**, (2021).
2. Hasan F, Chiu Y, Shaw RM, Wang J, Yee C. Hypoxia acts as an environmental cue for the human tissue-resident memory T cell differentiation program. *JCI Insight* **6**, (2021).
3. Sukumar M, et al. Inhibiting glycolytic metabolism enhances CD8⁺ T cell memory and antitumor function. *J Clin Invest* **123**, 4479-4488 (2013).
4. Shi LZ, et al. Interdependent IL-7 and IFN-gamma signalling in T-cell controls tumour eradication by combined alpha-CTLA-4+alpha-PD-1 therapy. *Nat Commun* **7**, 12335 (2016).
5. Wang R, et al. The transcription factor *Myc* controls metabolic reprogramming upon T lymphocyte activation. *Immunity* **35**, 871-882 (2011).
6. Peng M, Yin N, Chhangawala S, Xu K, Leslie CS, Li MO. Aerobic glycolysis promotes T helper 1 cell differentiation through an epigenetic mechanism. *Science* **354**, 481-484 (2016).
7. Dang EV, et al. Control of T(H)17/T(reg) balance by hypoxia-inducible factor 1. *Cell* **146**, 772-784 (2011).
8. Shi LZ, et al. HIF1alpha-dependent glycolytic pathway orchestrates a metabolic checkpoint for the differentiation of TH17 and Treg cells. *J Exp Med* **208**, 1367-1376 (2011).
9. Coyle AJ, et al. The CD28-related molecule ICOS is required for effective T cell-dependent immune responses. *Immunity* **13**, 95-105 (2000).
10. Yang Y, Chang JF, Parnes JR, Fathman CG. T cell receptor (TCR) engagement leads to activation-induced splicing of tumor necrosis factor (TNF) nuclear pre-mRNA. *J Exp Med* **188**, 247-254 (1998).

11. *Ajouaou Y, et al. The oxygen sensor prolyl hydroxylase domain 2 regulates the in vivo suppressive capacity of regulatory T cells. Elife 11, (2022).*
12. *Doedens AL, et al. Hypoxia-inducible factors enhance the effector responses of CD8(+) T cells to persistent antigen. Nat Immunol 14, 1173-1182 (2013).*
13. *Palazon A, et al. An HIF-1alpha/VEGF-A Axis in Cytotoxic T Cells Regulates Tumor Progression. Cancer Cell 32, 669-683 e665 (2017).*
14. *Kim WY, et al. Failure to prolyl hydroxylate hypoxia-inducible factor alpha phenocopies VHL inactivation in vivo. Embo j 25, 4650-4662 (2006).*
15. *Karagiannis F, et al. Impaired ketogenesis ties metabolism to T cell dysfunction in COVID-19. Nature 609, 801-807 (2022).*

REVIEWERS' COMMENTS

Reviewer #1 (Remarks to the Author):

This revised manuscript by Shen et al. has been improved, and the authors have responded well to my comments. Overall, the revised version of the manuscript is more robust, and the data support the findings. I have no major concerns remaining.

However, I would suggest including a bar graph to quantify the western blot data to better represent and interpret the results.

We have included bar graphs to quantify all the Western blot data.

Reviewer #2 (Remarks to the Author):

The authors have sufficiently addressed the main and specific points raised in the initial review and have satisfactorily answered pertinent questions and included new data that makes the manuscript more complete. The new data which addresses potential involvement of Hif2a in IFN γ production is acceptable and sufficient and inclusion of the data points taken at 2.5% oxygen makes the conclusions drawn by the authors stronger.

General comments: Authors need to avoid making sweeping conclusions within the discussion section of the paper that Hif1a is a metabolic regulator of IFN production in all hypoxic T cells as the data presented in this manuscript only indicates such regulation might be relevant in naïve T cells.

Also, authors should clarify the nomenclature used for mouse strains in the text with reference to the conditional knockouts.

Finally, please reduce the number of acronyms, especially in the discussion section, to make the text more readable to a broader audience.

Thanks for the points. We have indicated that our study is on IFN- γ induction in naïve T cells activated under hypoxia. We have clarified the nomenclature for mouse strains. We have limited acronyms in the discussion section, which have been defined when they appear first in the text.

Reviewer #3 (Remarks to the Author):

I appreciate that the authors addressed my two main concerns.

The addition of the stable isotope data and extracellular flux analysis data really adds to what was originally mostly expression data.

The epigenetic data also adds to the proposed mechanism and strengthens the paper.

We thank this reviewer for the great suggestions, which substantially improve our study.